# Learning to Drive with Two Minds: A Competitive Dual-Policy Approach in Latent World Models

## Abstract

End-to-end autonomous driving models trained solely with imitation learning (IL) often suffer from poor generalization. In contrast, reinforcement learning (RL) promotes exploration through reward maximization but faces challenges such as sample inefficiency and unstable convergence. A natural solution is to combine IL and RL. Moving beyond the conventional two-stage paradigm (IL pretraining followed by RL fine-tuning), we propose CoDrive, a competitive dual-policy framework that enables IL and RL agents to interact during training. CoDrive introduces a competition-based mechanism that facilitates knowledge exchange while preventing gradient conflicts. Experiments on the nuScenes dataset show an 18% reduction in collision rate compared to baselines, along with stronger generalization and improved performance on long-tail scenarios. Code is available at: `https://anonymous.4open.science/r/drive-with-two-minds`.

## 1 Introduction

End-to-end learning has become the mainstream paradigm in autonomous driving (Hu et al., 2023; Jiang et al., 2023; Weng et al., 2024). Unlike modular pipelines, end-to-end models allow gradients to propagate across perception, prediction, and planning, enabling all components to be optimized toward the final driving objective.

Most existing approaches rely on imitation learning (IL), where models are trained to mimic expert demonstrations. In practice, these methods are essentially supervised learning (SL): the model's outputs are directly supervised to match expert trajectories. The effectiveness of SL relies on the assumption that data are independent and identically distributed (IID). However, in embodied tasks such as driving, this assumption fails—observations are temporally correlated, and small prediction errors can accumulate, pushing the vehicle outside its "safety zone" and leading to cascading failures. As a result, IL (or more precisely, SL-based IL) agents often generalize poorly and struggle on long-tail scenarios.

To mitigate these issues, prior work has attempted to expand the training distribution, for example using generative world models (Wang et al., 2024; Wen et al., 2024; Gao et al., 2024). However, generated data remain limited in realism and computationally costly. RL offers another solution by encouraging exploration and learning from trial-and-error. Yet applying RL in simulators suffers from two drawbacks: (1) high-fidelity expert demonstrations are often unavailable. In fact, many "expert drivers" in simulators are themselves trained using RL rather than real-world data. Without genuine expert demonstrations, IL cannot be applied, which also prevents combining IL and RL in such settings. (2) Agents trained in simulators also face sim-to-real transfer challenges, where policies that succeed in virtual environments may fail in the real world. So in this paper, we investigate offline RL directly on expert datasets. While appealingly simple, this setting introduces new challenges: 1) Since experts already achieve near-optimal rewards, naively fitting expert transitions reduces RL to IL, offering limited exposure to novel states; 2) Non-reactive simulation can provide evaluation metrics (e.g., L2 error, collision rate) for hypothetical actions, but cannot yield new states following those actions.

To solve the two problems above, we 1) inspired by GRPO (Shao et al., 2024), we use group sampling, allowing the agent to generate multiple candidate action sequences and evaluate them through

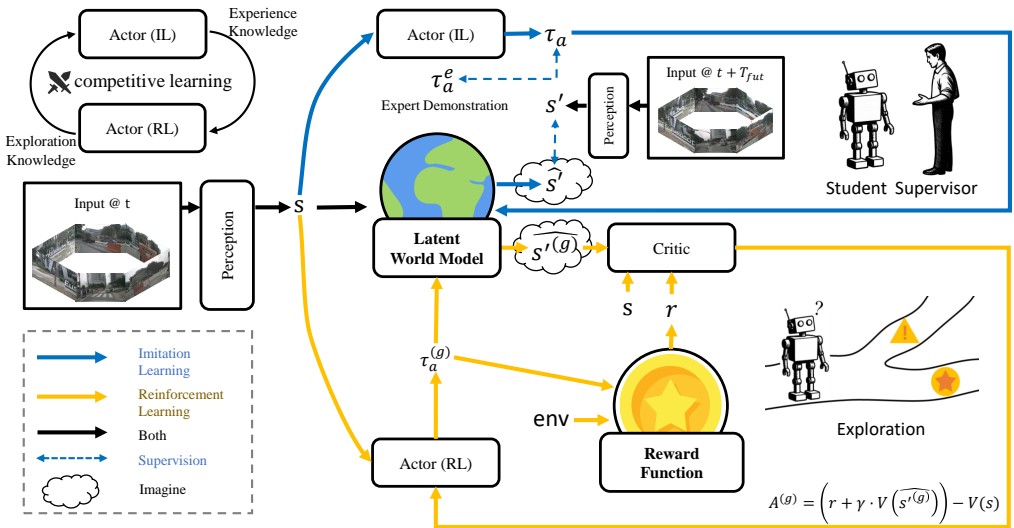

Figure 1: **Overview of CoDrive.** CoDrive adopts a dual-policy architecture that integrates imitation learning (IL) and reinforcement learning (RL) through a shared latent world model. In each iteration, the IL actor and RL actor are trained in parallel. The latent world model is learned during the IL phase and then used in the RL phase, where only the RL actor and critic are updated. For exploration, the RL actor samples multiple action sequences, predicts future states via the latent world model, and evaluates them with rule-based reward functions. The critic assigns advantages to each sequence based on the imagined trajectories and rewards. To promote interaction, a competitive learning mechanism exchanges knowledge between the IL and RL actors.

non-reactive simulation; 2) leverage a latent world model as a reactive simulator to predict future states conditioned on sampled actions, enabling imagination-based training beyond ground-truth data.

Finally, to integrate IL and RL without gradient interference (if simply add the loss of IL and RL), we introduce a dual-policy architecture that decouples the two objectives into separate actors. A competition-based learning mechanism fosters interaction and selective knowledge transfer between the IL and RL agents.

**Our contributions are summarized as follows:**

- We fully utilize the latent world model, which not only enhances the predictive capability of the IL Actor but also provides a lightweight internal simulator for the RL Actor to explore, thereby avoiding reliance on external simulators.

- We propose a dual-policy competitive learning framework that jointly trains IL and RL while encouraging interaction through structured competition.

- We conduct extensive experiments on nuScenes and Navsim, showing that our method improves generalization, reduces collisions, and achieves stronger performance on long-tail scenarios compared to baselines.

## 2 RELATED WORK

### 2.1 END-TO-END AUTONOMOUS DRIVING

End-to-end autonomous driving methods replaced traditional modular design by the end-to-end manner. UniAD (Hu et al., 2023) first demonstrated the potential of end-to-end by unifying perception and planning within a unified framework. VAD (Jiang et al., 2023) vectorized the scene representation and improved the efficiency of inference. PARA-Drive (Weng et al., 2024) decomposed the traditional pipeline and searched for optimal architectures. Some works also integrated

world model. LAW (Li et al., 2025b) and World4Drive (Zheng et al., 2025) predicted future visual latents via a world model, improving temporal understanding. SSR (Li & Cui, 2025) utilized sparse tokens to represent dense BEV features and similarly employed a world model to predict the next feature to enhance scene comprehension. WoTE (Li et al., 2025c) leverages a world model to predict future states, enabling online trajectory evaluation and selection. Our approach leverages world model as an offline simulator. Specifically, the RL policy iteratively interacts with world model to imagine future scene transitions, enabling reward-driven policy optimization.

## 2.2 RL IN AUTONOMOUS DRIVING

Reinforcement learning plays an important role in autonomous driving. Roach (Zhang et al., 2021) trained an RL expert to map BEV input to actions, subsequently served as teacher for the student model. VLM-RL (Huang et al., 2024) leveraged a Vision-Language-Model (VLM) to generate rewards signals for RL. Think2Drive (Li et al., 2024a) integrated DreamV3 to train an expert model, becoming the first agent to finish CARLA v2. AdaWM (Wang et al., 2025) analyzed performance degradation of driving agents, proposing a strategy that selectively updates actor or world model. Imagine2Drive (Garg & Krishna, 2025) proposed a novel framework by integrating a video world model (Gao et al., 2024) with a diffusion-based policy, achieving impressive performance in the CARLA. Urban Driver (Scheel et al., 2022) builds a differentiable simulator via perception outputs and high-fidelity HD maps, supporting efficient policy learning. Inspired by such approaches, our method performs RL within a latent space using a learned world model and integrates it with IL to achieve more stable and efficient training.

## 2.3 COMBINE IL AND RL IN AUTONOMOUS DRIVING

The combination of RL and IL is an important problem in autonomous driving. AutoVLA (Zhou et al., 2025) conducted supervised fine-tuning to learn how to reason and later applied GRPO (Shao et al., 2024) to achieve faster reasoning. RAD (Gao et al., 2025) constructed a large 3D environment and mixed IL and RL during training. TrajHF (Li et al., 2025a) used IL fine-tuning and RLHF on a large collected preference data and achieve impressive performance. ReCogDrive (Li et al., 2025d) incorporated expert imitation loss and RL-loss in simulator to explore safer trajectories. Our approach performed a competitive framework that optimize IL and RL simultaneously, allowing them to share information for safer action.

# 3 METHOD

## 3.1 ACTOR MODELING

Given current observation $o$ (usually images captured by cameras), the perception module encodes it into latent state $s \in \mathbb{R}^{B \times N_t \times D}$, where $B$ is the bach size, $N_t$ is the number of tokens for each latent state, and $D$ is the feature dimension. For planning, a waypoint query $Q_w \in \mathbb{R}^{B \times n \times D}$ is employed to extract the waypoint features $s_w = \{s_{w,1}, s_{w,2}, ..., s_{w,n}\} \in \mathbb{R}^{B \times n \times D}$ through cross-attention, where $n$ denotes the number of waypoints in a trajectory. The planning head then decodes the waypoint features $s_w$ into an action sequence $\tau_a = \{a_1, a_2, ..., a_n\}$. Using the provided expert action demonstrations $\tau_a^e$ as labels, imitation learning applies an L1 loss $L_{imi}$ to supervise output.

$$s_w = \{s_{w,1}, s_{w,2}, ..., s_{w,n}\} = \text{CrossAttn}(q = Q_w, k = s, v = s), \quad (1)$$

$$\tau_a = \{a_1, a_2, ..., a_n\} = \text{PlanningHead}(s_w), \quad (2)$$

$$L_{imi} = ||\tau_a - \tau_a^e||. \quad (3)$$

To further endow the model with the predictive ability, a world model is used to predict future states. Unlike pixel-level generative world models, we operate in latent space to reduce task complexity. Specifically, given the current state $s$ and action $\tau_a$, the world model predicts future state $\hat{s}'$, and the latent world model is trained in a self-supervised manner using mean square error (MSE), and we denote this loss $L_{wm}$:

$$\hat{s}' = \text{LatentWorldModel}(s, \tau_a). \quad (4)$$

Meanwhile, the perception module encodes next observation $o'$ into ground-truth state $s'$. The latent world model is trained in a self-supervise manner using mean square error (MSE)[1]. The overall imitation learning loss $L_{IL}$ combines $L_{wm}$ and $L_{imi}$, where $\alpha$ is a hyperparameter:

$$L_{wm} = \text{MSELoss}(s', \hat{s}'), \tag{5}$$

$$L_{IL} = L_{imi} + \alpha \cdot L_{wm}. \tag{6}$$

## 3.2 BACKWARD PLANNING

In practice, the planning head predicts $\tau_a$ in a single forward. Such design overlooks dependencies among each step in $\tau_a$. A natural extension adopts a self-attention layer with causal mask to introduce temporal causality. The policy for $a_i$ is formulated as $\pi_i(a_i|s_{w,1}, ..., s_{w,i}) = \pi(a_i|s_{w,j\leq i})$.

While this forward-causal design appears intuitive, human driving behavior suggests an alternative perspective. Drivers typically decide **where to go** before committing to low-level actions. Moreover, in real-world deployment, only the first action is executed before replanning, making earlier actions matter more.

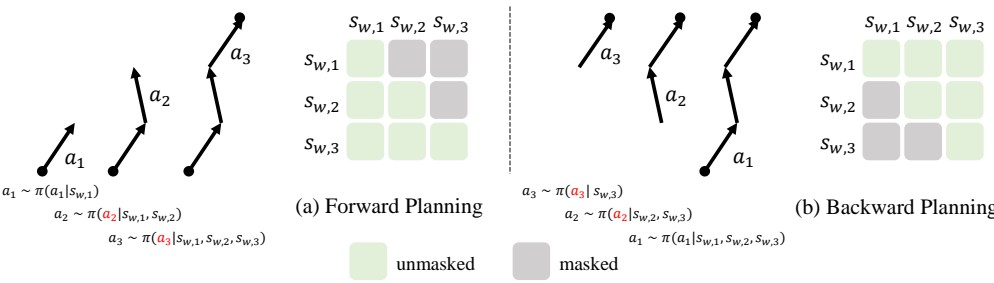

(a) Forward Planning      (b) Backward Planning

unmasked      masked

Figure 2: The comparison of forward planning (using causal mask) and backward attention (use inverse causal mask)

Motivated by these insights, as shown in Fig. 2, we explore a counterintuitive alternative– **backward planning** (inverse causality)–where the i-th action is conditioned on the current and future waypoint features:

$$\pi_i(a_i|s_{w,i}, ..., s_{w,n}) = \pi_i(a_i|s_{w,j\geq i}) \tag{7}$$

This formulation incorporates early actions with richer contextual information while leaving later actions less constrained, aligning better with how humans plan. Plus, inverse causality changes only the conditioning order, without affecting the smoothness of the final trajectory. Prior evidence (Liu et al., 2025) from embodied AI further supports this "goal-to-action" reasoning paradigm, and our experiments in Tab. 4 confirm that it consistently outperforms both forward-causal and non-causal baselines.

## 3.3 REINFORCEMENT LEARNING

RL needs rewards to evaluate the quality of explored trajectories. Given a predicted action sequence $\tau_a$, the corresponding position sequence is obtained via cumulative summation: $\tau_{pos} = \{a_1, a_1 + a_2, ..., \sum_i a_i\}$. On nuScenes, we use two components to define the reward: imitation reward $r_{imi}$ and collision reward $r_{col}$:

$$r_{imi}^{(i)} = e^{-||a_i - a_i^e||_2}, \quad r_{col}^{(i)} = 1 - \text{CollisionDetection}(\tau_{pos}, \text{env}). \tag{8}$$

Here, the env denotes static map information and dynamic agents' moving trajectories. The final reward for step $i$ is defined as:

$$r_i = r_{col}^{(i)} \cdot r_{imi}^{(i)}. \tag{9}$$

---

[1]In implementation, we follow LAW's setting, the next observation is actually the observation of future 1.5 seconds.

On Navsim (Dauner et al., 2024) benchmark, the reward is computed similarly and utilize the PDMS score to evaluate the quality of trajectories.

The deterministic action sequence produced by our model cannot be directly optimize with RL, which typically requires probabilistic policies. To address this, we need to model the uncertainty of the action sequence. Similar with the planning head output the mean value $\mu_i$ for each action, we use a stochastic head to output uncertainty for each action:

$$\tau_\sigma = \{\sigma_1, \sigma_2, ..., \sigma_n\} = \text{StochasticHead}(s_w), \tag{10}$$

where $\sigma_i$ is the standard deviation for $a_i$. Here we simply use Gaussian distribution to model each action and use the diagonal matrix to serve as the covariance matrix. Then the policy for action $a_i$ can formulated with:

$$\pi_i(a_i|s_{w,j\geq i}) = \mathcal{N}(\mu_i, \sigma_i^2 I), \tag{11}$$

where $I$ denotes the identity matrix. Given the offline imitation dataset $\{(s, \tau_a^e, \tau_r^e, s'), ...\}$, using Gaussian Log Likelihood loss can easily fitting and cloning the behavior of experts': $L_{bc} = -\sum_{i=1}^n \log^{\pi_i(a_i^e|s_{w,j\geq i})}$. But we wish the RL Actor can explore and learn from both good examples but also bad examples. Inspired by GRPO (Shao et al., 2024), we introduce exploration by sampling $G$ trajectories from the policy and use the rule-based reward function to calculate its corresponding reward sequences. Each action sequence in the group, together with its corresponding reward sequence, can be formally expressed as:

$$\tau_i^{(g)} = \{a_1^{(g)} \sim \pi_1(a_1^{(g)}|s_{w,j\geq 1}), ..., a_n^{(g)} \sim \pi_n(a_n^{(g)}|s_{w,j\geq n})\}, \tau_r^{(g)} = \{r_1^{(g)}, r_2^{(g)}, ..., r_n^{(g)}\}. \tag{12}$$

For each trajectory, we compute its total reward and normalize within the group using Z-score normalization to obtain the advantage and the naive policy gradient loss with group sampling (naive PGGS) is computed by:

$$L_{actor} = -\frac{1}{G}\sum_{g=1}^G A^{(g)} \cdot (\sum_{i=1}^n \log \pi_i(a_i^{(g)}|s_{w,j\geq i})), A^{(g)} = \frac{\Sigma\tau_r^{(g)} - \text{mean}(\Sigma\tau_r^{(1)}, ..., \Sigma\tau_r^{(G)})}{\text{std}(\Sigma\tau_r^{(1)}, ..., \Sigma\tau_r^{(G)})}. \tag{13}$$

To extend the advantage estimation to long-term rewards, we train a critic model $V$ to output the value of both current state $s$ and the next state $s'$. Since the offline dataset does not provide next states for sampled trajectories, we leverage the latent world model to generate rollouts:

$$\hat{s'}^{(g)} = \text{LatentWorldModel}(s, \tau_a^{(g)}). \tag{14}$$

The long-term advantage $A_{long}^{(g)}$ is computed by:

$$A_{long}^{(g)} = (\sum r^{(g)} + \gamma \cdot V(\hat{s'}^{(g)})) - V(s), \tag{15}$$

in which $\gamma$ is a discount factor. We further apply Z-score normalization to $A_{long}^{(g)}$ within each group and denote the normalized advantage as the critic advantage $A_{cri}^{(g)}$. During training, the actor and critic are then jointly optimized, and this is the method of actor + dreaming critic with group sampling (ADCGS):

$$L_{act} = -\frac{1}{G}\sum_{g=1}^G A_{cri}^{(g)} \cdot (\sum_{i=1}^n \log \pi_i(a_i^{(g)}|s_{w,j\geq i})), \quad L_{cri} = \frac{1}{G}\sum_{g=1}^G [V(s) - (\sum \tau_r^{(g)} + \gamma \cdot V(\hat{s'}^{(g)}))]^2. \tag{16}$$

Since pure RL with our reward is hard to converge (see results in Tab. 5), To stabilize training and guide the RL actor toward human-like exploration, we incorporate a small behavior cloning term, $L_{bc}$, with a small coefficient $\beta = 0.005$. This auxiliary loss provides weak expert guidance during the competitive phase, where the RL actor would otherwise operate without access to expert supervision.

$$L_{RL} = L_{act} + L_{cri} + \beta \cdot L_{bc}. \tag{17}$$

In practice, since actions in a sampled sequence are drawn independently from different policies, the resulting position trajectory $\tau_{pos}$ may lack smoothness. To address this, we adopt a step-aware mechanism: within each sampled sequence, only one action is stochastic, while the remaining actions are

set to the mode of their respective policies, ensuring a smoother $\tau_{pos}$. The detailed algorithm and visualizations are provided in Appendix A.1. To further stabilize critic learning, we employ the two-critic trick, where a reference critic maintains an exponential moving average (EMA) of the learning critic.

### 3.4 DUAL-POLICY LEARNING FRAMEWORK

During training, the model's planning module is decoupled into IL actor and RL actor, optimized by $L_{IL}$ and $L_{RL}$ respectively. To encourage the two actor interact with each other, two actor can compete and share information with each other.

To balance the contributions of the imitation learning (IL) actor and the reinforcement learning (RL) actor, we periodically compare their performance every k iterations. The comparison is based on the difference of the accumulative reward scores achieved by IL Actor and RL Actor ($\Delta r_{acc}$), along with two threshold ($\lambda_{min}$, $\lambda_{max}$) to measure the size of the score difference, and a hyperparameter $p$ to control the knowledge swap degree.

Based on the score difference $\Delta r_{acc}$, we adaptively update the loser actor's parameters: 1) If the two actors perform similarly, we keep both unchanged; 2) If the performance gap is moderate, we apply soft merging to gradually transfer knowledge from the winner to the loser; 3) If the gap is large, we directly replace the loser's parameters with the winner's. And we can formulated with the following equation:

$$
loser.weight := \begin{cases} loser.weight, & \text{if } \lambda_{min} \geq |\Delta r_{acc}| \\ loser.weight \cdot p + winner.weight \cdot (1-p), & \text{if } \lambda_{min} < |\Delta r_{acc}| \leq \lambda_{max} \\ winner.weight. & \text{if } |\Delta r_{acc}| > \lambda_{max} \end{cases} \quad (18)
$$

This adaptive mechanism enables stable cooperation between IL and RL actors, preventing premature dominance while allowing faster convergence once one actor becomes consistently superior.

## 4 EXPERIMENT

### 4.1 BENCHMARKS

**nuScenes (Caesar et al., 2020)** is a large-scale autonomous driving benchmark featuring 1,000 20-second urban driving scenes with 1.4M annotated 3D boxes across 23 object classes. It provides 360° imagery from six cameras and 2Hz keyframe annotations. Following prior works (Hu et al., 2023; Jiang et al., 2023), we evaluate planning using L2 placement error and Collision Rate.

**Navsim (Dauner et al., 2024)** is a compact, filtered version of OpenScenes (Contributors, 2023), itself derived from nuPlan (Karnchanachari et al., 2024). It emphasizes challenging scenarios and contains 120 hours of driving at 2Hz. It includes *navtrain* and *navtest* splits for training and testing. To better reflect closed-loop safety and behavior, Navsim evaluates agents with six metrics: No at-fault Collisions (NC), Drivable Area Compliance (DAC), Time to Collision (TTC), Ego Progress (EP), Comfort (C), and Driving Direction Compliance (DDC). These are combined into a weighted Driving Score (PDMS).

### 4.2 IMPLEMENTATION DETAILS

For experiments on nuScenes, our method builds upon the LAW framework (Li et al., 2025b) and SSR framework (Li & Cui, 2025). We train our models using 8 NVIDIA A800-SXM4-80GB GPUs and perform evaluation on the same A800 GPU. The training is conducted with a batch size of 1 using the AdamW optimizer, with a learning rate set to $5 \times 10^{-5}$, we use cosine annealing learning rate with linear warm up. All other training settings follow the original LAW and SSR configuration. The training process takes approximately 20 hours to complete.

For experiments on Navsim, we adopt the Transfuser (Prakash et al., 2021) model as backbone. Transfuser employs a Transformer-based architecture to fuse front-view camera image and LIDAR data across multiple stages. We train our model on navtrain split and evaluate it on test split, using

the same hardware configuration as in nuScenes experiments. The training is performed with a batch size of 16 using the AdamW optimizer, with a learning rate set to $1 \times 10^{-4}$.

## 4.3 MAIN RESULTS

The results on nuScenes are presented in Tab. 1. For nuScenes, We follow the evaluation protocol of (Jiang et al., 2023), which reports average L2 distance and collision rate over 1s, 2s, and 3s prediction horizons. We tried our method on both SSR (Li & Cui, 2025) and LAW (Li et al., 2025b), and we found that SSR is unstable, even using the same random seed. So in ablation studies, we only use LAW.

Table 1: **Comparison of state-of-the-art methods on the nuScenes dataset.** *All models are trained and evaluated on 8 A800 GPUs. †SSR exhibits instability on our machine, so results are reported using random seed 0. ‡The perception-free version of LAW does not support temporal augmentation, so temporal augmentation is enabled here for fair comparison with other methods. The best results are shown in **bold**, and the second-best results are underlined.

| Method | L2 (m) ↓ | | | | Col (%) ↓ | | | | L2 · Col ↓ |
|---|---|---|---|---|---|---|---|---|---|
| | 1s | 2s | 3s | Avg. | 1s | 2s | 3s | Avg. | Avg. |
| **Perception-based Methods** | | | | | | | | | |
| ST-P3 | 1.33 | 2.11 | 2.90 | 2.11 | 0.23 | 0.62 | 1.27 | 0.71 | 1.50 |
| UniAD | 0.48 | 0.96 | 1.65 | 1.03 | 0.05 | 0.17 | 0.71 | 0.31 | 0.32 |
| VAD | 0.41 | 0.70 | 1.05 | 0.72 | 0.07 | 0.17 | 0.41 | 0.22 | 0.16 |
| PARA-Drive | 0.25 | 0.46 | 0.74 | 0.48 | 0.14 | 0.23 | 0.39 | 0.25 | 0.12 |
| GenAD | 0.28 | 0.49 | 0.78 | 0.52 | 0.08 | 0.14 | 0.34 | 0.19 | 0.10 |
| LAW (perception-based) | 0.24 | 0.46 | 0.76 | 0.49 | 0.08 | 0.10 | 0.39 | 0.19 | 0.09 |
| **Perception-free Methods** | | | | | | | | | |
| BEV-Planner | 0.30 | 0.52 | 0.83 | 0.55 | 0.10 | 0.37 | 1.30 | 0.59 | 0.32 |
| SSR*† | 0.18 | 0.35 | 0.62 | **0.38** | 0.48 | 0.45 | 0.51 | 0.48 | 0.18 |
| LAW* | 0.32 | 0.62 | 1.03 | 0.66 | 0.08 | 0.13 | 0.46 | 0.22 | 0.15 |
| LAW+CoDrive* | 0.29 | 0.59 | 1.00 | 0.63 | 0.06 | 0.10 | 0.37 | 0.18 | 0.11 |
| LAW+CoDrive*‡ | 0.23 | 0.42 | 0.70 | 0.45 | 0.10 | 0.12 | 0.30 | **0.17** | **0.08** |

On nuScenes, LAW+CoDrive consistently outperforms the baseline LAW in both L2 error and collision rate. Even without temporal augmentation (commonly used by other methods), LAW+CoDrive already achieves the best collision rate. After enabling temporal augmentation, its L2 error further decreases substantially. Overall, our method attains the best performance on the combined L2 · Col metric, demonstrating its effectiveness. Moreover, our approach shares the same inference architecture as the baseline and introduces no additional latency. Further details are provided in Appendix A.2.1 and A.2.2.

**generalization ability and Long-tail scenario performance on nuScenes**

We evaluate cross-city generalization by training on nuScenes-Singapore and testing on nuScenes-Boston (Tab. 2 and Fig. 3a). The original LAW shows poor generalization when trained in one city and evaluated in another, whereas our method, benefiting from the RL actor's exploration, demonstrates markedly improved robustness.

Table 2: **Cross-city generalization ability test on nuScenes dataset.**

| Method | L2 (m) ↓ | | | | Col (%) ↓ | | | | L2 · Col ↓ |
|---|---|---|---|---|---|---|---|---|---|
| | 1s | 2s | 3s | Avg. | 1s | 2s | 3s | Avg. | Avg. |
| LAW | 0.45 | 0.89 | 1.46 | 0.93 | 0.13 | 0.43 | 1.50 | 0.69 | 0.64 |
| LAW+CoDrive | 0.33 | 0.65 | 1.13 | **0.70** | 0.04 | 0.15 | 0.46 | **0.22** | **0.15** (↓ 77%) |

To assess long-tail performance, we construct two subsets from the evaluation set: one with high L2 error and one with high collision rate, identified using the baseline model. Results (Fig. 3b) show that our method improves both generalization and long-tail robustness. Details on subset construction and full results are in Appendix A.2.3.

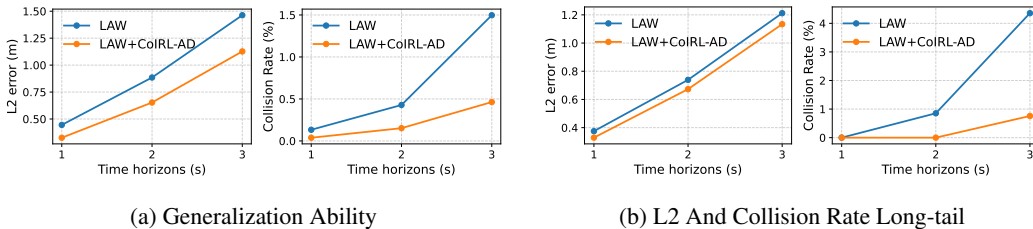

(a) Generalization Ability          (b) L2 And Collision Rate Long-tail

Figure 3: Comparision of Generalization and Performance on Long-tail Dataset (L2 and Collision).

The results on Navsim are shown in Tab. 3. For Navsim, we adopt the close-loop metrics provided in Navsim. Specifically, we use the test split rather than navtest split for evaluation, as the former contains much more scenarios (5044) than the latter (885), making it more suitable for comprehensively assessing the model's overall driving performance.

Table 3: **Comparison of state-of-art methods on Navsim test set.** *reproduced by us. † test on navtest set.

| Method | NC ↑ | DAC ↑ | TTC ↑ | Comf.↑ | EP ↑ | PDMS↑ |
|---|---|---|---|---|---|---|
| Human | 100.0 | 100.0 | 100.0 | 99.9 | 87.5 | 94.8 |
| Ego Status MLP | 93.0 | 77.3 | 83.6 | 100.0 | 62.8 | 65.6 |
| VADv2 (Chen et al., 2024) | 97.2 | 89.1 | 91.6 | 100.0 | 76.0 | 80.9 |
| UniAD (Hu et al., 2023) | 97.8 | 91.9 | 92.9 | 100.0 | 78.8 | 83.4 |
| PARA-Drive (Weng et al., 2024) | 97.9 | 92.4 | 93.0 | 99.8 | 79.3 | 84.0 |
| Transfuser (Prakash et al., 2021) | 97.7 | 92.8 | 92.8 | 100.0 | 79.2 | 84.0 |
| LAW (Li et al., 2025b) | 96.5 | 95.4 | 88.7 | 99.9 | 81.7 | 84.6 |
| Hydra-MDP (Li et al., 2024b) | 98.3 | 96.0 | 94.6 | 100.0 | 78.7 | 86.5 |
| WoTE* (Li et al., 2025c) | 98.6 | 96.4 | 95.3 | 100.0 | 81.1 | 87.9 |
| WoTE+CoDrive | **98.6** | **96.8** | **95.5** | **100.0** | 81.0 | **88.2** |

On navsim, our model obtains a PDMS of 88.2, outperforming recent state-of-the-art methods, showing notable improvements across multiple sub-metrics, including NC (+0.4), DAC (+0.4) and TTC (+0.8). Compared to WoTE (Li et al., 2025c), which leverages a world model to evaluate candidate trajectories during testing, our approach achieves a higher overall score.

## 4.4 ABLATION STUDY

**Causality**     To verify the effect of inverse causality, we conduct three experiments on LAW + CoDrive naive PGGS model, and change the mask we use in the self attention layer to $s_w$ in three different ways: 1) no causal mask; 2) causal mask[2]; 3) inverse causal mask[3]. We set $\beta$ in $L_{RL}$ equals to 0. The results is shown in Tab. 4.

Table 4: The Effect of Causality to the Performance

| Method | L2 (m) ↓ | | | | Collision Rate (%) ↓ | | | | L2 · Col ↓ |
|---|---|---|---|---|---|---|---|---|---|
| | 1s | 2s | 3s | Avg. | 1s | 2s | 3s | Avg. | Avg. |
| LAW | 0.32 | 0.63 | 1.03 | 0.66 | 0.09 | 0.12 | 0.46 | 0.22 | 0.15 |
| no mask | 0.30 | 0.60 | 1.04 | **0.64** | 0.09 | 0.15 | 0.43 | 0.22 | 0.14 |
| causal mask | 0.35 | 0.67 | 1.10 | 0.71 | 0.08 | 0.16 | 0.57 | 0.27 | 0.19 |
| causal mask (inv) | 0.31 | 0.61 | 1.01 | 0.65 | 0.04 | 0.08 | 0.48 | **0.20** | **0.13** |

From the results, the naive causal mask increases both L2 error and collision rate. In contrast, removing the mask or using the inverse causal mask outperforms the baseline. The no-mask setting

---

[2]causal mask: *torch.triu(torch.ones(n,n), diagonal=1).bool()*
[3]inverse causal mask: *torch.tril(torch.ones(n,n), diagonal=-1).bool()*

reduces L2 error, while the inverse causal mask improves both L2 and collision rate, highlighting the effectiveness of backward planning (inverse causality).

**Integration of IL and RL.** We compare several integration strategies: (i) *loss merging*, jointly optimizing with $L_{IL} + L_{RL}$; (ii) *IL–RL interval*, alternating between $L_{IL}$ and $L_{RL}$; (iii) *two-stage*, pre-training with $L_{IL}$ then fine-tuning with $L_{RL}$; and (iv) *decoupled actors*, where IL and RL actors are optimized separately, optionally with competition ("comp"). Results are shown in Tab. 5.

Table 5: The Performance of Different Ways to Integrate IL and RL.

| Description | L2 (m) ↓ | | | | Collision Rate (%) ↓ | | | | L2 · Col |
|---|---|---|---|---|---|---|---|---|---|
| | 1s | 2s | 3s | Avg. | 1s | 2s | 3s | Avg. | Avg. |
| pure IL | 0.32 | 0.63 | 1.03 | 0.66 | 0.09 | 0.12 | 0.46 | 0.22 | 0.15 |
| pure RL | 3.92 | 6.55 | 9.18 | 6.55 | 2.75 | 4.87 | 7.72 | 4.93 | 32.29 |
| loss merging | 0.38 | 0.73 | 1.17 | 0.76 | 0.03 | 0.12 | 0.54 | 0.23 | 0.17 |
| IL-RL interval | 0.31 | 0.63 | 1.07 | 0.68 | 0.12 | 0.17 | 0.54 | 0.28 | 0.19 |
| two-stage | 2.43 | 4.21 | 6.03 | 4.22 | 2.29 | 4.13 | 6.53 | 4.32 | 18.23 |
| decouple, w/o comp | 0.32 | 0.64 | 1.07 | 0.68 | 0.09 | 0.13 | 0.53 | 0.25 | 0.17 |
| decouple, w/ comp | 0.31 | 0.61 | 1.01 | **0.65** | 0.04 | 0.08 | 0.48 | **0.20** | **0.13** |

From Tab. 5, only the *decouple, w/ comp* variant improves both L2 and collision rates over the baseline. This is notable since two-stage IL–RL transfer is effective in other domains (e.g., Deepseek's R1 (Guo et al., 2025)). We attribute the limited gains to: (1) overly simple rewards (imitation and collision only), (2) use of a basic actor–critic method instead of more stable algorithms like PPO, and (3) non-reactive simulation, where both states and rewards are generated by the world model, introducing bias. These factors explain the poor "pure RL" results and the degradation in most RL-augmented variants. Nevertheless, the competitive decoupled design demonstrates that effective IL–RL interaction can still yield measurable improvements.

## 5 ANALYSIS

### 5.1 COMPETITION ANALYSIS

The last two lines of result in Tab. 5 show that the competitive learning mechanism can help the IL Actor and RL Actor interact and finally learn a better model, but how?

By tracking metrics such as accumulated wins and score differences (IL score – RL score) over iterations, we observe the following: (1) In the early stage (<20k iterations), the IL actor achieves more wins and higher scores, indicating that IL initially leads the learning process. Because initially, the latent world model provides no meaningful signal, leaving the RL actor unable to extract any useful guidance from exploration or interaction. (2) Afterward, as the latent world model starts to encode the underlying driving dynamics, the RL actor progressively learns fundamental driving behaviors through its competitive interplay with the IL actor. And RL Actor's exploration via group sampling becomes more effective than simply imitating expert trajectories. Consequently, RL achieves higher scores and dominates in later training. This progression resembles the two-stage paradigm (IL pretraining followed by RL fine-tuning), but with a key difference: IL and RL are trained jointly. Even though IL loses more frequently in later stages, its gradients continue to benefit shared components such as the perception module.

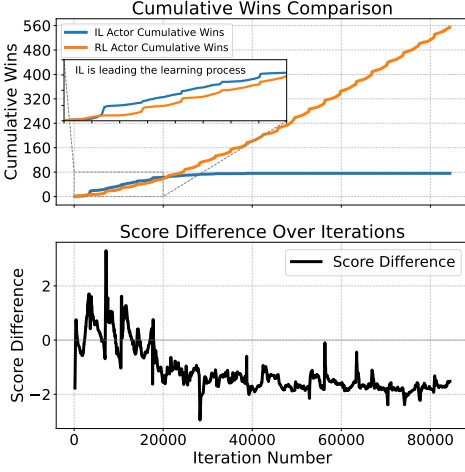

Figure 4: Accumulated wins (top) and score difference (bottom) across training iterations.

## 5.2 QUALITATIVE ANALYSIS

We further visualize the planned trajectories on nuScenes to qualitatively compare our method with the baseline LAW. The results show that, with reinforcement learning and self-exploration, our model is better at recognizing hazardous situations and proactively avoiding potential collisions with surrounding vehicles and pedestrians. In contrast, the baseline LAW often fails to react appropriately, likely due to its reliance on imitating expert trajectories without understanding the underlying intent. Additional qualitative examples are provided in Appendix A.3.

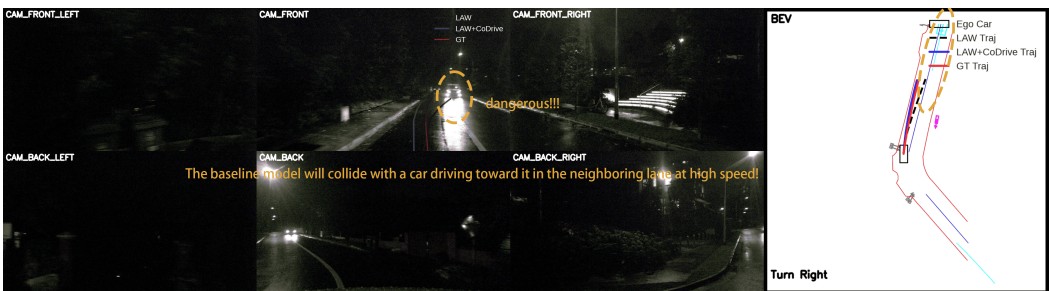

Figure 5: The baseline model collides with a fast-approaching vehicle in the adjacent lane, whereas our model successfully avoids the collision.

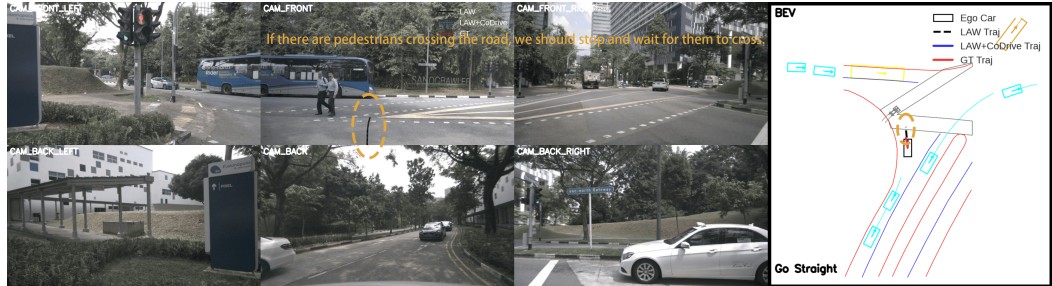

Figure 6: When pedestrians are crossing ahead, our model stops and yields, while the baseline model continues moving forward.

## 6 CONCLUSION

We presented a competitive dual-policy framework that integrates IL and RL for end-to-end autonomous driving. Motivated by IL's limitations in generalization and long-tail performance, we exploit RL's exploration capability within an offline setting. By combining group sampling with non-reactive simulation and augmenting it with imagination via a latent world model, we train an RL actor capable of capturing long-term advantages beyond immediate rewards. A competition-based mechanism further enables effective interaction between IL and RL actors to promoting knowledge sharing. Experiments on nuScenes and Navsim demonstrate that our approach significantly reduces collisions, improves generalization, and enhances long-tail performance. We believe this framework provides a promising direction for combining imitation and reinforcement learning in embodied AI, and we hope it inspires future research in autonomous driving and beyond.

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

## A  APPENDIX

### A.1  STEP-AWARE REINFORCEMENT LEARNING

When the RL Actor explore, it samples action sequence from $n$ policies (see Eq. 12). The problem is, as we model each action in the action sequence separately, when sample an action sequence, the actions are actually sampled independently from different Gaussian distribution. When we use the sampled action sequence $\tau_a^{(g)}$ to calculate the position sequence $\tau_{pos}^{(g)}$, the trajectory is unstable and not smooth. That means in exploration, the RL Actor's driving trajectories is possibly do not satisfy some kinematic characteristics. Below we visualize the comparison of naive group sampling and our step aware method: Fig. 7 (straight), Fig. 8 (left turn), Fig. 9 (right turn). Our goal is to let the RL Actor output reasonable driving trajectories, and simply use group sampling here is inefficient (there is no need to explore some unreasonable trajectories), so in implementation we use step aware mechanism.

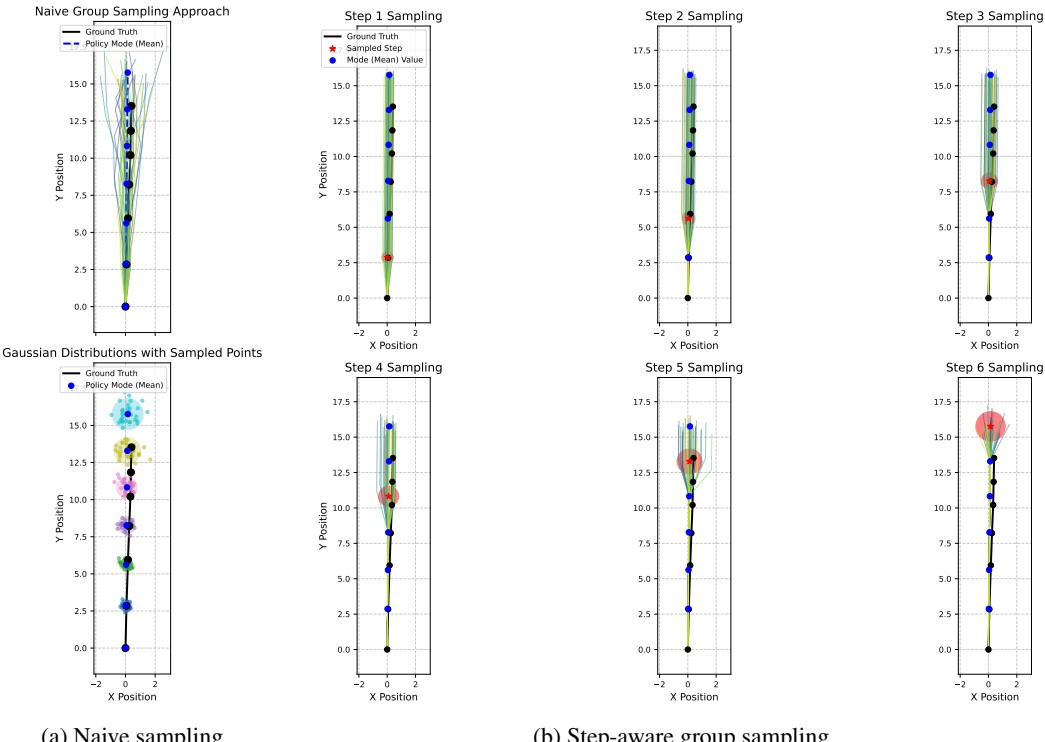

(a) Naive sampling            (b) Step-aware group sampling

Figure 7: Comparison between naive group sampling method and our step-aware group sampling method in straight driving situation (in training process)

More specifically, via group sampling, we actually get $G$ samples for each action in the action sequence. The idea is, we decoupled the exploration in to each step, i.e. we do not sample $n$ actions from different policies, we ensure that only one action will be sampled in each exploration, and for the resting $n-1$ actions in the same exploration, we simply use the mode action (the expectation $E[\pi_i]$ in gaussian actually). The process is visualized in Fig below. We formulate our methods in Alo 1.

### A.2  MORE EXPERIMENT RESULTS

#### A.2.1  TRAINING TIME

Our method introduces additional training cost due to the use of reinforcement learning, the RL actor's exploration, and the proposed competitive mechanism. The training time comparison is provided in Table 6. Although the training time roughly doubles, we consider this overhead ac-

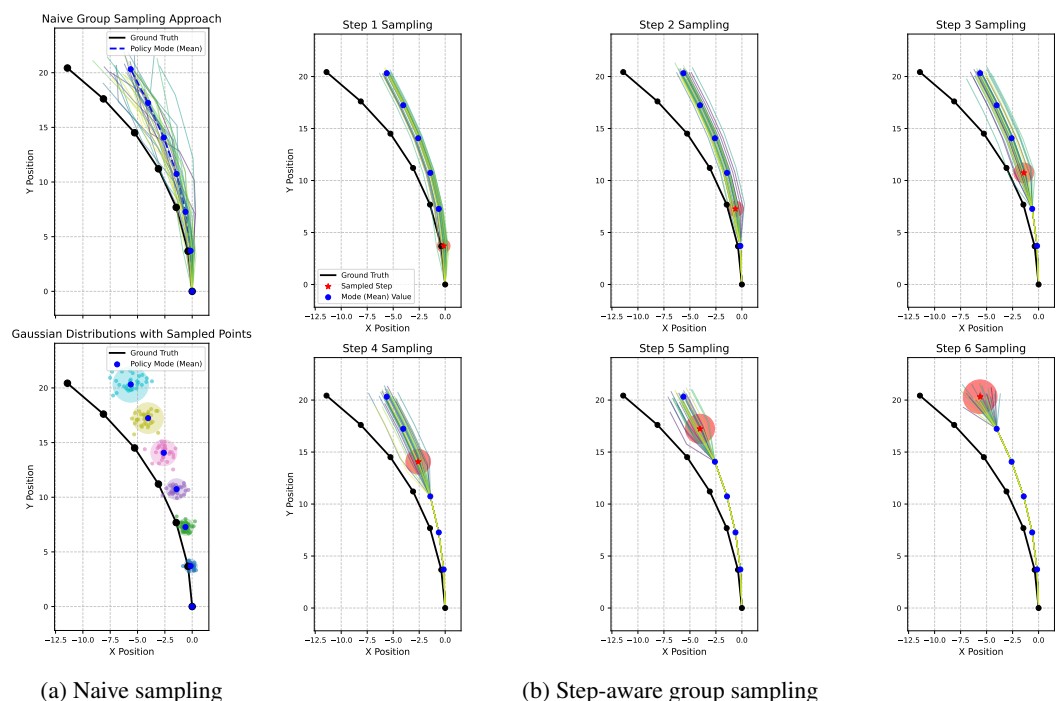

(a) Naive sampling       (b) Step-aware group sampling

Figure 8: Comparison between naive group sampling method and our step-aware group sampling method in left turn driving situation (in training process)

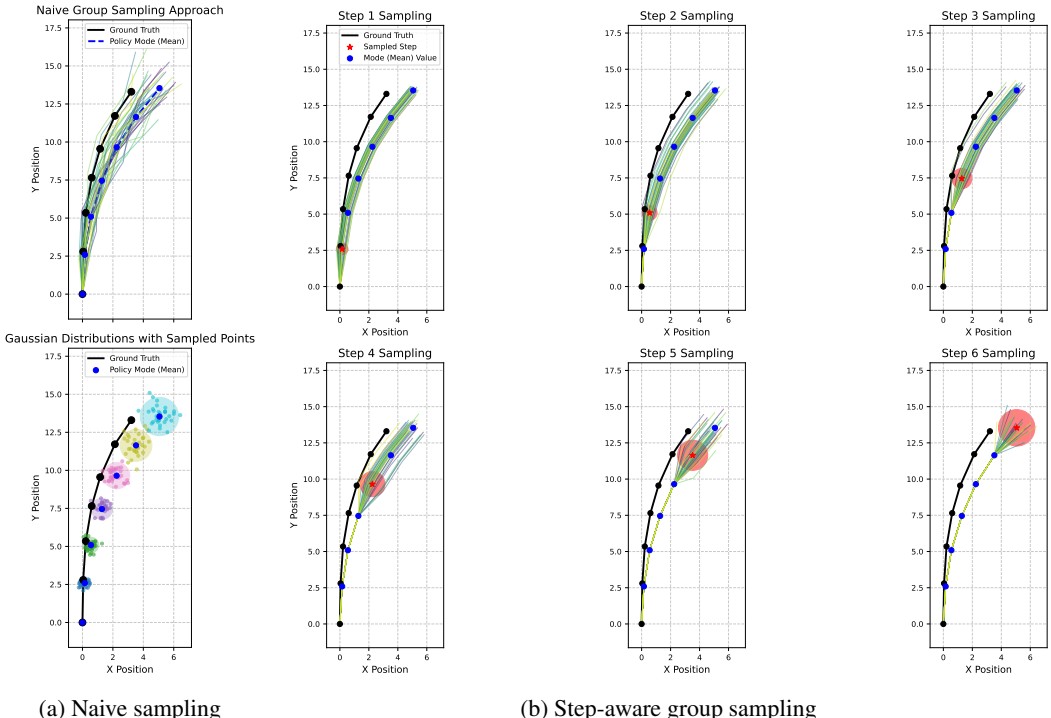

(a) Naive sampling       (b) Step-aware group sampling

Figure 9: Comparison between naive group sampling method and our step-aware group sampling method in right turn driving situation (in training process)

---

**Algorithm 1** Step Aware RL with Group Sampling

---

**Input:** $\{\pi_1, \pi_2, ..., \pi_n\}$, $s$, $s_w$, $V_\psi$ (Critic Model), $(i, g, L_{actor}, L_{critic} \leftarrow 0)$

1: **repeat**
2:    $i \leftarrow i + 1$
3:    **repeat**
4:      $g \leftarrow g + 1$
5:      $\tau_a^{(g)} \leftarrow \{E[\pi_1], ..., a_i^{(g)} \sim \pi_i(a_i^{(g)}|s_{w,j \geq i}), ..., E[\pi_n]\}$
6:      Calculating reward $\tau_r^{(g)}$ based on Eq.8, 9
7:      Predict future state $\hat{s'}^{(g)}$ based on Eq. 14
8:      Computing "long-term" advantage $A_{long}^{(g)}$ based on Eq. 15
9:    **until** $g = G$
10:    Computing critic advantage for step i, $A_{critic} = \text{Z-Score-Norm}(\{A_{long}^{(1)}, ..., A_{long}^{(G)}\})$
11:    $L_{actor} \leftarrow L_{actor} - \frac{1}{G} \sum_{g=1}^{G} A_{critic}^{(g)} \cdot \left( \sum_{j=1}^{n} \log^{\pi(\tau_a^g[j]|s_{w,k \geq j})} \right)$
12:    $L_{critic} \leftarrow L_{critic} + \frac{1}{G} \sum_{g=1}^{G} \left[ V_\psi(s) - \left( \sum \tau_r^{(g)} + \gamma \cdot V_\psi(\hat{s'}(g)) \right) \right]^2$
13: **until** $i = n$
14: $L_{actor} \leftarrow \frac{1}{n} \cdot L_{actor}$
15: $L_{critic} \leftarrow \frac{1}{n} \cdot L_{critic}$
**Output:** Loss of actor $L_{actor}$ and dreaming critic $L_{critic}$

---

ceptable. Most RL-based methods require long training schedules to converge, and in our case, the additional cost leads to a substantial improvement in generalization while leaving inference latency unchanged—a crucial property for embodied tasks such as autonomous driving.

Table 6: Training time on the nuScenes training set.

| Method | Training Epochs | GPU Usage | Time |
|---|---|---|---|
| LAW | 24 | 8×A800-80G | 10 h |
| LAW+CoDrive | 24 | 8×A800-80G | 20 h |

A.2.2 INFERENCE TIME

During inference time, our method don't introduce extra inference time cost. Specifically, the knowledge of RL actor had shard with IL actor and they can forward at the inference time. We tested the inference metrics of LAW and our method on a single A100, and the results is shown in Tab. 7

Table 7: Inference time

| Method | fps | latency (ms) |
|---|---|---|
| LAW | 26.99 | 37.1 |
| LAW+CoDrive | 27.1 | 37.0 |

A.2.3 GENERALIZATION AND PERFORMANCE ON LONG-TAIL SCENARIOS

**Details on Long-tail Subset Construction** We define long-tail scenarios according to two criteria: high L2 prediction errors and high collision rates, and accordingly construct two specialized long-tail datasets. The L2 Long-Tail Dataset is built by first selecting scenes with `fut_valid_flag=TRUE`, and then filtering for scenes with L2 distance greater than 0.3 at 1s, greater than 0.5 at 2s, and simultaneously greater than 1.0 at 3s. This results in a total of 984 scenes for testing. The Collision Rate Long-Tail Dataset is obtained by selecting scenes with `fut_valid_flag=TRUE` and excluding all scenes with a zero collision rate at the 3s horizon, yielding 91 test scenes.

**Detailed Results**   The detailed results of performance on two long-tail subset are shown in Tab. 8 and Tab. 9.

Table 8: Long-tail dataset (L2) comparision results

| Method | L2 (m) ↓ | | | | Collision Rate (%) ↓ | | | |
|---|---|---|---|---|---|---|---|---|
| | 1s | 2s | 3s | Avg. | 1s | 2s | 3s | Avg. |
| LAW | 0.375 | 0.739 | 1.212 | 0.775 | 0 | 0.080 | 0.459 | 0.180 |
| LAW+CoDrive | **0.329** | **0.673** | **1.134** | **0.712** | 0 | **0.053** | **0.335** | **0.129** |

Table 9: Long-tail dataset (Collision) comparison results

| Method | L2 (m) ↓ | | | | Collision Rate (%) ↓ | | | |
|---|---|---|---|---|---|---|---|---|
| | 1s | 2s | 3s | Avg. | 1s | 2s | 3s | Avg. |
| LAW | 0.336 | 0.669 | 1.184 | 0.740 | 0 | 0.852 | 4.356 | 1.736 |
| LAW+CoDrive | **0.283** | **0.577** | **0.987** | **0.616** | 0 | **0** | **0.758** | **0.253** |

### A.3   MORE QUALITATIVE RESULTS

In this section, we provide additional qualitative results to further demonstrate the effectiveness of our approach.

### A.3.1   GOOD CASES

From the visualization results, we observe that after incorporating reinforcement learning, our model successfully avoids many collisions that the baseline model fails to handle. This reveals that integrating IL and RL is an effective design choice.

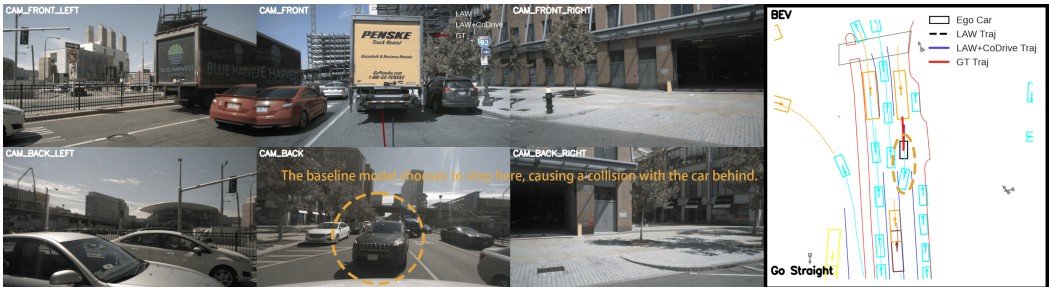

Figure 10: Good Case: In a queuing scenario while waiting for the green light, our model proceeds after the car ahead turns off its indicator and starts moving, whereas the baseline model remains stopped, which may lead to a rear-end collision.

### A.3.2   BAD CASES

However, there are also some scenarios where both the baseline model and our model perform unsatisfactorily.

### A.4   USAGE OF LLMS

During our research, we used LLMs in the following ways:

- **Text polishing.** We employed LLMs to refine the writing style of our drafts and to shorten sections when the main text exceeded the page limit (9 pages). We did not use LLMs to generate new content; all polished text was manually reviewed before inclusion in the submission.

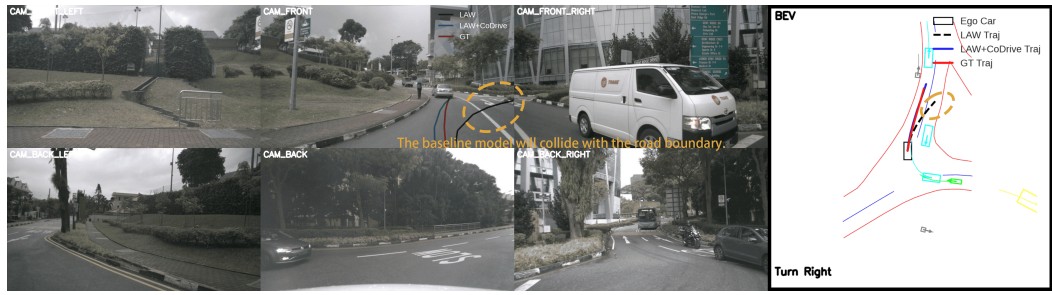

Figure 11: Good Case: When the high-level command is incorrect, our model adapts its planned trajectory based on the driving scenario, whereas the baseline model fails to adjust.

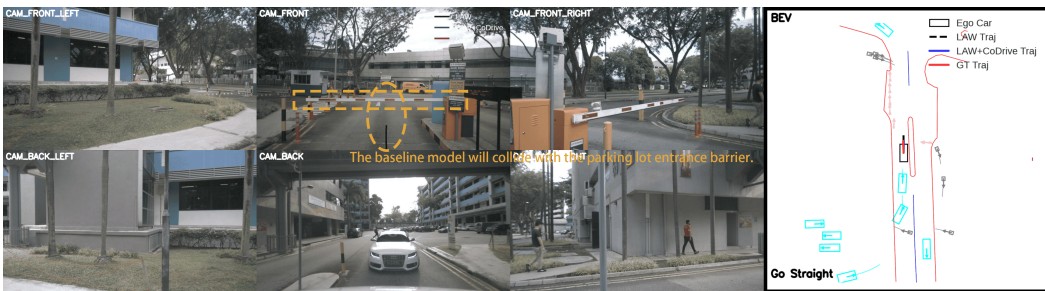

Figure 12: Good Case: When exiting the parking lot, our model waits for the toll barrier to rise, whereas the baseline model proceeds forward and collides with it.

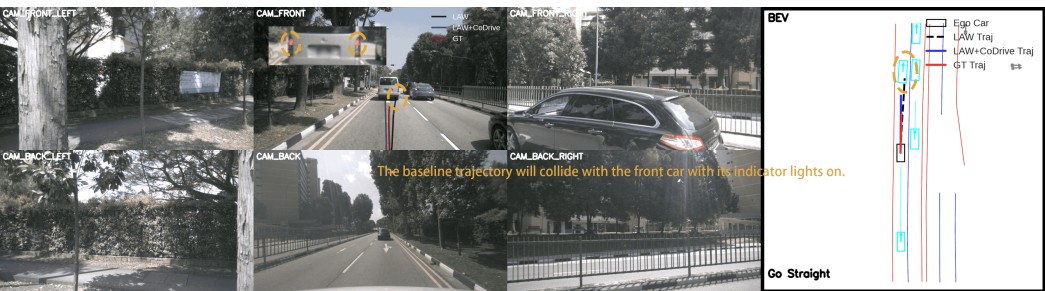

Figure 13: Good Case: When the leading vehicle has its indicator on, our model slows down to avoid a rear-end collision, whereas the baseline model maintains speed and causes a collision.

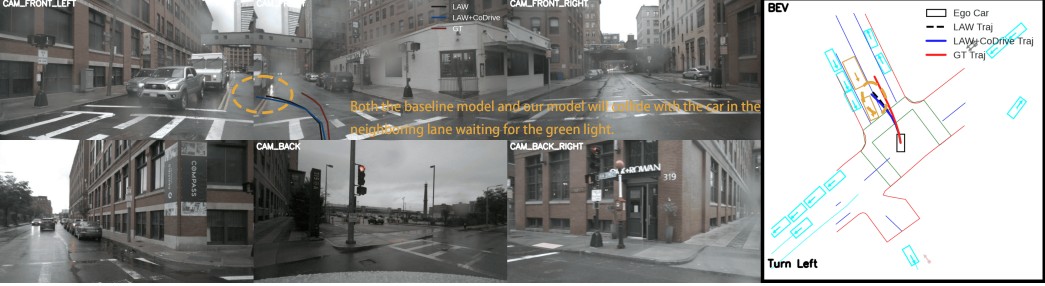

Figure 14: Bad Case: When turning right at an intersection, both the baseline model and our model plan trajectories that collide with a vehicle waiting for the green light.

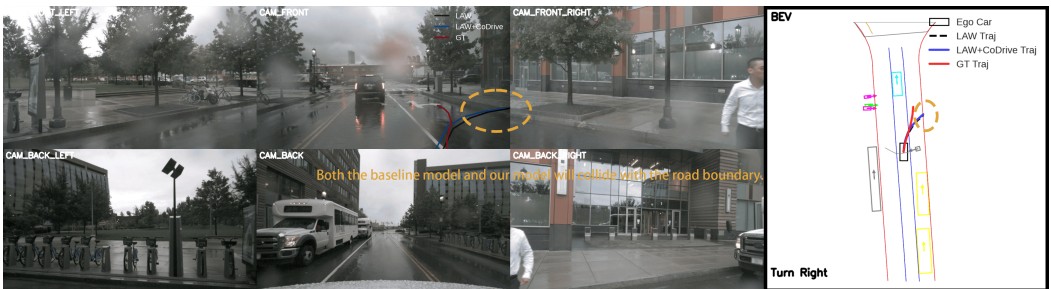

Figure 15: Bad Case: When pedestrians are crossing, our model continues to move forward at a low speed instead of stopping completely.

Figure 16: Bad Case: Both the baseline model and our model fail to plan reasonable trajectories in lane-changing scenarios.

- **Idea exploration.** We used LLMs as a tool for brainstorming and literature search assistance. By discussing our ideas with LLMs, we were directed toward relevant research areas and key related works, which we then examined ourselves.

- **Code assistance.** LLMs were used to help debug programs by analyzing error logs, to review our code for potential issues, and to generate auxiliary visualization scripts. Except for visualization, we did not rely on LLMs to produce experimental code. For visualization, we first drafted data-loading code ourselves, then refined the visualization with LLM-generated snippets, carefully verifying correctness before use.

