# OpenReview forum: "Learning to Drive with Two Minds: A Competitive Dual-Policy Approach in Latent World Models"
_ICLR.cc/2026/Conference — Submitted to ICLR 2026_

### Official Review · Reviewer_T6eo · 2025-10-21

**Soundness:** 2
**Presentation:** 3
**Contribution:** 2
**Rating:** 4
**Confidence:** 5

**Summary:**

The paper "Learning to Drive with Two Minds: A Competitive Dual-Policy Approach in Latent World Models" proposes a dual-policy learning framework that trains an Imitation Learning (IL) actor and a Reinforcement Learning (RL) actor in parallel. The two actors periodically merge their model parameters based on predefined merging strategies and performance comparison.

**Strengths:**

1. The idea of maintaining both IL and RL actors throughout training, rather than discarding IL after warm-up, is interesting and could inspire further exploration.
2. The "backward planning" concept of providing more contextual information to early actions is intuitively appealing.
3. The paper is clearly motivated and connects well with the general problem of combining imitation and reinforcement learning.

**Weaknesses:**

1. The claimed contribution, "We integrate RL into an end-to-end driving framework by leveraging a latent world model for imagination-based simulation, avoiding reliance on external simulators," is not new. Previous work has explored similar ideas (see [1] for example).
2. The notations in the paper are sometimes confusing. For instance, what is the dimension of the waypoint features s_w? The symbols
s and s_t are not used consistently. tau_a is an action sequence but is sometimes referred to as a single action (Equation (4)). L_{wm} appears before being defined. The notations for action sampling in Figure 2 are incorrect. The format of the L_{bc} is also incorrect.
3. While the dual-actor training pipeline introduces additional training overhead, the corresponding performance improvement appears minor.

Reference:
[1] Scheel, Oliver, et al. "Urban driver: Learning to drive from real-world demonstrations using policy gradients." Conference on Robot Learning. PMLR, 2022.

**Questions:**

1. The backward planning approach actually breaks the MDP assumption in RL. How can use this backward planning in RL? Also, what is the horizon length for the future states?
2. The latent world model is used during the RL actor learning phase, but when is this world model trained?
3. Is there any particular reason for using the L1 loss for imitation learning in Equation (3)?

---

> ### Author Response · Authors · 2025-11-13
> **Response to Reviewer T6eo's Valuable Feedback (part-1).**
>
> ## W1: One of the contributions is not original.
>
> Thank you for pointing this out.
> We agree that our original phrasing might have overstated the novelty. In the revised version, we will restate our first contribution as follows:
> > “We fully utilize the latent world model, which not only enhances the predictive capability of the IL Actor but also provides a lightweight internal simulator for the RL Actor to explore, thereby avoiding reliance on external simulators.”
>
> We also plan to revise Section 2.2 (RL in Autonomous Driving) to better position our work within prior research. In particular, we will add the following discussion referencing the suggested work:
> > “Urban Driver [1] builds a differentiable simulator via perception outputs and high-fidelity HD maps, supporting efficient policy learning. Inspired by such approaches, our method performs RL within a latent space using a learned world model and integrates it with IL to achieve more stable and efficient training.”
>
> These revisions will appear in our updated manuscript. Once the revision is ready, we will inform you here.
>
> ## W2: Notations in the paper are sometimes confusing.
> Thank you very much for your careful and detailed review. We address each notation issue below:
> 1. **Dimension of the waypoint features $s_w$.**
>
> In Equation 1:
> $s_w $= {$s_{w, 1}, s_{w, 2}, ..., s_{w, n} $} = $\text{CrossAttn}(q=Q_w, k=s, v=s),$
> Where $Q_w \in R^{B \times n \times D}$ and $s \in R^{B \times N_t \times D}$. Here $n$ is the number of waypoints in a trajectory, $D$ is hidden feature dimension, and $N_t$ is the number of tokens for each state.
>
> 2. **The symbols $s$ and $s_t$ are not used consistently.**
>
> You are correct—there is a misusage of $s_t$ in equation (4): $\hat{s'} = \text{LatentWorldModel}(s_t, \tau_a)$. We will correct this inconsistency in our revision. Other parts of the paper already use $s$ consistently.
>
> 3. **Clarification of $\tau_a$.**
>
> $\tau_a$ denotes **an action sequence, not a single action**. In Equation (4), we treat the entire action sequence as a single input entity to the latent world model.
>
> 4. **$L_{wm}$ appears before definition.**
>
> We will revise the relevant section to read:
> > "The latent world model is trained in a self-supervised manner using mean square error (MSE), and we denote this loss $L_{wm}$".
>
> 5. **Incorrect notations for action sampling in Figure 2.**
>
> Thank you for catching this—we will fix the notation accordingly in the revised figure.
>
> 6. **Incorrect format of $L_{bc}$.**
>
> We will revise the text before Equation (17) to:
> > “We add a small behavior cloning term $L_{bc}$ to stabilize training.”
>
> We sincerely appreciate your detailed feedback and will make all these corrections in our revision. Once ready, we will inform you here.
>
> ## W3: Performance improvement appears minor.
>
> Thank you for pointing that out!
>
> As summarized in Table 1, our method achieves lower L2 error (↓0.66 → 0.63, −4.5%) and collision rate (↓0.22% → 0.18%, −18%) compared to the baseline LAW. While the overall gains may appear moderate, we emphasize that our method brings **consistent improvements in generalization and long-tail robustness**, which are crucial in autonomous driving.
> - **Cross-city generalization**:
> In Figure 4(a) and Table 6, our model surpasses LAW with a **33% reduction in L2 error** (1.24 → 0.83) and a **72% reduction in collision rate** (1.09% → 0.30%).
> - **Long-tail scenarios**:
>   - Long-tail L2 scenes (Figure 4(b), Table 7): **L2 ↓10% (0.78 → 0.70)**, **Collision ↓22% (0.18% → 0.14%)**.
>   - Long-tail collision scenes (Figure 4(b), Table 8): **L2 ↓7% (0.73 → 0.68)**, **Collision ↓33% (1.74% → 1.17%)**.
>
> These results demonstrate that our dual-policy framework substantially enhances **robustness and generalization**.

---

> ### Author Response · Authors · 2025-11-13
> **Response to Reviewer T6eo's Valuable Feedback (part-2).**
>
> ## Q1-1: The backward planning approach actually breaks the MDP assumption in RL.
>
> We appreciate this insightful question.
> Indeed, classical RL relies on the Markov assumption, where $s_{t+1}$ is independent with $s_{t-1}$, given $s_t$.
>
> However, in our setting, the **RL actor operates on an action sequence** $\tau_a$={$a_1, a_2, ... a_n$} as a **single structured action**, rather than modeling stepwise transitions $(s_1, a_1, s_2, a_2, ..., s_n)$.
>
> Therefore, our training objective concerns transitions of the form $(s, \tau_a, r, s')$, where the backward planning strategy only influences **how the current action sequence is generated**, not the transition dynamics themselves.
>
> So, even though we use backward planning, we only **change the way to output the action at current state**, which is independent with future state and the latent world model still predicts the next latent state $s'$ based solely on the current state $s$ and the sequence $\tau_a$ . That's why we do not violate the markov assumption here.
>
> Additionally, we experimented with a strictly causal variant (i.e., using a causal mask instead of backward planning), and the performance degraded — **L2 error increased from 0.66 → 0.71**, and **collision rate from 0.22% → 0.27%**, further confirming the practical value of backward planning (see our Table 3).
>
> ## Q1-2: what is the horizon length for the future states?
>
> We follow the setting of LAW [2], using a **1.5-second horizon**, which was empirically shown to perform best among 0.5 s, 3 s, and 10 s horizons (see Table 6 in [2]).
>
> For consistency and simplicity, we also adopt LAW’s latent world model design, where the **entire action sequence** is input into the model to predict the future state. Although this formulation may appear unconventional, it has proven effective in both LAW [2] and our experiments.
>
> ## Q2: when is this world model trained?
>
> As mentioned in **W2-4**, the **latent world model** is optimized jointly within the **imitation learning phase**, as part of the overall imitation loss. Once trained, this world model is reused during **reinforcement learning** to provide imagination-based rollouts for the RL actor. Thus, it is **trained in the imitation loop** and **utilized in the reinforcement loop** without additional supervision.
>
> ## Q3: Any particular reason for using the L1 loss for imitation learning in Equation (3)?
>
> We follow prior end-to-end autonomous driving frameworks such as LAW [2] (Equation 6) and SSR [3] (Equation 9), both of which adopt the **L1 loss** for its simplicity and stable convergence behavior. Our choice is therefore consistent with these widely used baselines rather than introducing a new formulation.
>
> ## Reference
>
> [1] Scheel, Oliver, et al. "Urban driver: Learning to drive from real-world demonstrations using policy gradients." Conference on Robot Learning. PMLR, 2022.
>
> [2] Li, Yingyan, et al. "Enhancing end-to-end autonomous driving with latent world model." ICLR, 2025.
>
> [3] Li, Peidong, and Dixiao Cui. "Navigation-guided sparse scene representation for end-to-end autonomous driving." ICLR, 2025.

---

> > ### Comment · Reviewer_T6eo · 2025-11-13
> >
> > Thanks for your quick response to my questions. Regarding Q2, my understanding is that imitation learning and reinforcement learning are trained jointly, rather than sequentially as described in your explanation. Please let me know if I’ve misunderstood anything.

---

> > > ### Author Response · Authors · 2025-11-14
> > > **Response to reviewer T6eo's questions**
> > >
> > > Thank you for your question!
> > >
> > > You understood it correctly — in our method, imitation learning (IL) and reinforcement learning (RL) are both active during training. Two loops run in parallel:
> > > - **Imitation Learning Loop**: trains the IL Actor using imitation loss, and simultaneously updates the latent world model through self-supervised learning.
> > > - **Reinforcement Learning Loop**: treats the latent world model as an online simulator, allowing the RL Actor to interact with it and be optimized via the policy gradient objective.
> > >
> > > In the baseline setting with a single actor, integrating IL and RL typically relies on one of two strategies:
> > > - **Loss merging**: combining IL and RL objectives into a single joint loss.
> > > - **Two-stage training**: first optimizing the actor with IL loss, then fine-tuning it with RL loss.
> > >
> > > Our design makes the two parallel loops possible because we **decouple the single actor into two separate actors**:
> > >
> > > an IL Actor optimized purely by the IL objective, and an RL Actor optimized purely by the RL objective.
> > >
> > > We hope this clarifies the reasoning behind our design choice.
> > >
> > > If you have any further questions, we’d be happy to discuss more!

---

> ### Author Response · Authors · 2025-11-14
> **We'd like to provide more information for reviewer T6eo**
>
> To further clarify the implementation details, we would also like to share our code. All identifying information has been removed, and the relevant file is available in an anonymous GitHub repository (filename is `LAW.py` because we build our method on LAW):
> https://anonymous.4open.science/r/drive-with-two-minds/core_code/LAW/LAW.py
>
> In the forward_pts_train function, you will find the two training loops in each training iteration:
> - **Lines 262–279: Imitation Learning Loop**
>   - The **IL Actor** corresponds to `pts_bbox_head`.
>   - `loss_rec` implements $L_{wm}$ from Equation (6).
>   - `loss_waypoint` implements $L_{imi}$ from Equation (6).
> - **Lines 281–336: Reinforcement Learning Loop**
>   - The **RL Actor** corresponds to `pts_bbox_head_rl`.
>   - `loss_rl` implements $L_{actor}$ from Equation (17).
>   - `loss_critic` implements $L_{critic}$ from Equation (17).
>   - `loss_rl_bc` implements $L_{bc}$ from Equation (17).
>
> The latent world model is trained in IL loop (line 266), and serves as an online simulator for RL Actor in RL loop (line 308).

---

> > ### Comment · Reviewer_T6eo · 2025-11-14
> >
> > Thanks for clarifying your training design. I’m asking because you mentioned that the latent world model is still being optimized during the imitation-learning phase, yet the reinforcement-learning phase (concurrently with IL) already relies on this latent model. Since the world model is not fully trained at that point, how does an evolving (i.e., still-under-training) latent world model reliably support the RL stage?

---

> ### Author Response · Authors · 2025-11-15
> **Response to reviewer T6eo's thoughtful questions**
>
> Thank you for your thoughtful question!
>
> You are absolutely right that, during training, the latent world model is still being optimized while the RL Actor is already using it for interaction. Below, we explain why this setup still works stably and why we intentionally chose it over a warm-up strategy.
>
> ## Why can RL start even when the world model is still evolving?
> Although all modules (IL Actor, latent world model, RL Actor) are trained from scratch, the early-stage instability of the world model does not destabilize training:
>
> **The world model is not affected by RL’s noisy interactions.**
> In early iterations, the RL Actor receives almost “noise-like” rollouts from the world model and therefore learns very little. However, this noisy exploration does **not** update the world model. The world model is optimized only through the IL loop (self-supervision + imitation), allowing it to steadily learn realistic driving dynamics.
>
> **The competitive mechanism provides a safety net for RL.**
> During early training, the IL Actor quickly improves by learning directly from the expert. In competitions, the IL Actor consistently obtains higher scores, so the RL Actor frequently adopts the IL Actor’s parameters. This achieves two things:
> - RL Actor is continuously “reset” to a sensible policy grounded in expert behavior.
> - By the time the world model becomes sufficiently accurate, RL already starts from an expert-like initialization and can meaningfully explore.
>
> **Empirical evidence from training dynamics.**
> As shown in Figure 5 of our paper, before ~20k iterations, IL Actor dominates because the world model is still maturing. After that, once the world model becomes reliable, the RL Actor, now equipped with both expert knowledge and a usable model, begins to score higher consistently.
>
> ***
>
> From here, we think you may raise a question:
> > So, it seems that the competitive mechanism is just provide some time for latent world model training and IL Actor learning. It seems easier to first train a latent world model, then directly use the RL Actor to explore it?
>
> For this question, below is our answers to why we use competitive mechanism:
>
> **IL Actor and world model are trained jointly.**
> In Equation (6), gradients of the world model flow through to the IL Actor. This coupling gives IL Actor predictive-awareness (i.e., “future-informed” representations). Training them separately breaks this synergy (see ablations in Table 5 of LAW [1]).
>
> **Two-stage training causes instability or collapse.**
> After training IL Actor + world model, there are two ways to proceed:
> - **Ignore IL Actor and train RL from scratch in the trained world model** → This corresponds to “pure RL” in Table 4, which collapses (L2 = 6.55, Collision = 4.93%). RL cannot converge without expert priors (at least cannot converge with the same epoch number that our method can converge).
> - **Initialize RL Actor from IL Actor and fine-tune via RL with the trained world model** → This corresponds to “two-stage” in Table 4. In this case, RL drifts away from expert behaviors without any constrain, suffers **catastrophic forgetting**, and still fails to converge within a reasonable number of epochs (L2 = 4.22, Collision = 4.32%).
>
> Thus, neither variant works well in our setting.
>
> ## Why competition helps (intuition)
>
> the competitive mechanism ensures that:
>
> - RL Actor **continuously receives expert knowledge**, preventing early collapse.
> - RL Actor **never drifts too far** from expert policy due to the competition constraint (only if RL Actor performs well, it can).
> - RL can gradually take over once the world model becomes accurate enough.
>
> In other words, competition smoothly bridges the gap between IL-driven learning and RL-driven exploration, avoiding catastrophic forgetting while allowing RL to shine when the world model is ready.
>
>
> We hope this explanation clarifies why our design does not require a separate warm-up phase and how the competitive mechanism stabilizes training. We are happy to discuss further if you have more questions!
>
>
> ## Reference
>
> [1] Li, Yingyan, et al. "Enhancing end-to-end autonomous driving with latent world model." ICLR, 2025.

---

> > ### Comment · Reviewer_T6eo · 2025-11-25
> >
> > Thank you for the authors’ thorough responses. I am satisfied that the majority of my concerns have been adequately addressed, and I have adjusted my score accordingly.

---

> > > ### Author Response · Authors · 2025-11-26
> > > **Thanks for reviewer T6eo's time and effort putting into reviewing our paper**
> > >
> > > Thank you for your kind words and for adjusting your score! We are delighted that our explanations have addressed most of your concerns. Your insightful suggestions, particularly regarding the formula details, competition mechanism, and warmup settings, have been invaluable in improving the quality of our work. We truly appreciate the time and effort you put into reviewing our paper, and we are grateful for your support. Thank you once again for your thoughtful feedback and for raising your score!

---

### Official Review · Reviewer_xpD5 · 2025-10-31

**Soundness:** 2
**Presentation:** 3
**Contribution:** 2
**Rating:** 2
**Confidence:** 4

**Summary:**

This paper addresses the limitations of end-to-end autonomous driving models, which suffer from poor generalization when trained with Imitation Learning (IL) and instability when trained with Reinforcement Learning (RL). The authors propose CoDrive, a novel dual-policy framework that synergistically combines IL and RL within a learned latent world model. Instead of relying on external simulators, CoDrive enables imagination-based training where an IL actor and an RL actor are trained in parallel. The core idea is a competitive learning mechanism that facilitates structured knowledge exchange: the IL actor provides expert knowledge, while the RL actor explores novel states and actions. This approach aims to leverage the stability of IL and the exploratory power of RL without their objectives directly conflicting, leading to improved generalization, reduction in collisions on the nuScenes dataset, and better performance in long-tail scenarios.

**Strengths:**

1. Novel and Well-Motivated Architecture for Integrating IL and RL. The proposed competitive learning mechanism provides a structured way to facilitate knowledge transfer.

2. This approach smartly bypasses the need for high-fidelity, hand-crafted external simulators, thereby mitigating the notorious sim-to-real gap and the dependency on expert demonstrations within the simulator.

3. By enabling imagination-based training, the framework allows the RL agent to perform sample-efficient and safe exploration of countless possible future scenarios.

**Weaknesses:**

1. In Table 1, the improvement gain on L2 metric is marginal. e.g. SSR+CoDrive is worse than SSR; LAW+CoDrive (PGGS) only achieves 0.01 gain. This raises questions about the practical significance of the proposed method, especially when weighed against its added architectural complexity.

2. In Table 2, the improvement on Navsim test set is also very minor. Taken together, this raises a critical question about the overall effectiveness of the proposed CoDrive framework,

3. How is the efficiency and resource-usage comparison? (Latency / training time / computation overheads)

4. In line 125: "achieve more stable training.". Any number comparison for proving the more stable training than single RL?

**Questions:**

See Weaknesses.

---

> ### Author Response · Authors · 2025-11-13
> **Response to Reviewer xpD5's Feedback**
>
> ## W1/W2: Improvement gain on metrics like L2 is marginal.
>
> Thank you for your feedback.
>
> Regarding the marginal L2 improvement of our model, we would like to clarify that even a small improvement in L2 is meaningful in our setting. Below are our reasons:
> 1. **L2 measures human-likeness.** The L2 metric reflects the similarity between the planned trajectories and those of human expert drivers. Since human behavior is difficult to represent with a single, simple reward function, it is inherently challenging for the RL actor to optimize its trajectory to be more human-like.
> 2. **Collision rate reflects safety.** Collision rate, on the other hand, can be clearly defined by a single reward function, making it easier for the RL actor to optimize.
> 3. **Balancing imitation and safety.** The goal of combining IL and RL is to maintain human-like behavior while improving safety. Our experiments show that our model achieves both — it significantly reduces the collision rate while keeping (and even slightly improving) the L2 error.
> 4. **On the apparent marginal improvement in Table 1.** The small L2 gain in Table 1 is mainly due to many simple scenarios in the evaluation set, where both our method and the baseline already perform well, leaving little room for improvement. To better highlight the strength of our approach, we focused on **high-L2 (long-tail)** scenarios based on the baseline LAW’s performance. As shown in Fig. 4(b) and Table 7, our model reduces the L2 error from **0.78 to 0.70** (a **10%** drop) in these challenging cases.
>
> As for your “critical question about the overall effectiveness of the proposed CoDrive framework,” we summarize our key findings as follows:
> 1. **Improved safety while maintaining human-likeness.** Compared with the baseline, our method achieves a lower collision rate while keeping L2 performance comparable or better.
> 2. **Enhanced generalization and long-tail robustness.** With the introduction of RL, our model demonstrates stronger generalization ability and improved performance in long-tail scenarios.
>
> **Supporting evidence**:
> - **Overall performance (Table 1)**:
>  L2 ↓ 0.66 → 0.63 (−4.5%), Collision ↓ 0.22% → 0.18% (−18%) vs. baseline LAW.
> - **Cross-city generalization (Figure 4(a), Table 6)**:
>  L2 ↓ 1.24 → 0.83 (−33%), Collision ↓ 1.09% → 0.30% (−72%).
> - **Long-tail scenarios (Figure 4(b), Table 7&8)**:
>   - Long-tail L2 scenes (Table 7): L2 ↓ 0.78 → 0.70 (−10%), Collision ↓ 0.18% → 0.14% (−22%).
>   - Long-tail collision scenes (Table 8): L2 ↓ 0.73 → 0.68 (−7%), Collision ↓ 1.74% → 1.17% (−33%).
>
> ## W3: Efficiency and resource-usage comparison
> Thank you for pointing that out!
> We evaluated inference efficiency in **Section A.2.1 (Inference Time)** of our appendix (Table 5). The detailed results tested on a single **A100 GPU** are as follows:
>
> | method | fps | latency (ms) |
> |--------------|----------|-------------|
> | LAW            | 26.99     | 37.1       |
> | LAW+CoDrive      | 27.1     | 37.0        |
>
> During inference, since the IL Actor and RL Actor have already exchanged knowledge during training, only the IL Actor is used for trajectory planning. Therefore, our framework introduces **no additional inference latency**.
>
> ## W4: More stable training than single RL.
> We compared the performance of **pure RL** versus our **IL+RL with latent world model** approach (see **Table 4**). The results clearly show that simple RL algorithms (e.g., policy gradient, as used in our paper) tend to **collapse and fail to converge**, yielding very poor performance (average L2 = **6.55**, Collision Rate = **4.93%**).
>
> In contrast, our proposed method (“decoupled, w/ comp”) achieves **stable and convergent training**, with an average **L2 = 0.65** and **Collision Rate = 0.20%**.
>
> These results quantitatively demonstrate that the combination of IL guidance and the latent world model effectively stabilizes RL training and prevents divergence.

---

> > ### Comment · Reviewer_xpD5 · 2025-11-13
> >
> > Thanks for your response!
> > How is the training time comparison of CoDrive with baselines?

---

> > > ### Author Response · Authors · 2025-11-13
> > > **Thanks for reviewer xpD5's response!**
> > >
> > > Thank you for your quick reply!
> > > Below is the comparison of **training time** between **CoDrive** and the baseline **LAW**. Both models were trained on the same server with **8×A800 GPUs** for **24 epochs** on the nuScenes training set.
> > >
> > > | method | training time (h) |
> > > |--------------|----------|
> > > | LAW            |  ~10   |
> > > | LAW+CoDrive      |  ~20   |
> > >
> > > Introducing the RL component and the competitive mechanism approximately doubles the training time, but yields clear improvements in both performance and generalization ability.
> > >
> > > To ensure transparency, we have uploaded the corresponding training logs (with all identifying information removed) to an **anonymous GitHub repository**:
> > > - LAW log: https://anonymous.4open.science/r/drive-with-two-minds/materials/logs/LAW_epoch_24.log
> > > - CoDrive log: https://anonymous.4open.science/r/drive-with-two-minds/materials/logs/CoDrive_epoch_24_final_version.log
> > >
> > > Thank you again for your engagement! We would be happy to further discuss any other questions or suggestions you may have.

---

> > > > ### Comment · Reviewer_xpD5 · 2025-11-27
> > > >
> > > > Thanks for your responses. Most of my concerns have been addressed. I recommend the authors to move the generalization experiments to the main paper part to highlight the effectiveness of the model. I've raised my score.

---

> > > > > ### Author Response · Authors · 2025-11-28
> > > > > **Thank you for your time and constructive feedback**
> > > > >
> > > > > We are very glad to hear that our responses addressed your concerns. Thank you for your thoughtful suggestion regarding the placement of the generalization experiments, we agree that highlighting these results in the main paper can better showcase the effectiveness of our method, and we will incorporate this improvement in our revision.
> > > > >
> > > > > We sincerely appreciate the time and effort you devoted to reviewing our work, as well as your score adjustment. Thank you again for your valuable feedback and support!

---

### Official Review · Reviewer_sWaj · 2025-10-31

**Soundness:** 3
**Presentation:** 3
**Contribution:** 2
**Rating:** 4
**Confidence:** 4

**Summary:**

The paper presents a policy framework that integrates an Imitation Learning and Reinforcement Learning approach to jointly learn a driving policy. For trajectory generation for autonomous driving, both learners predict actions but depending on the resulting score, only the best solution is implemented and weights of the worse learner are merged or replaced by the weights of the better learner. The approach is compared on the nuScenes and Navsim dataset against other baselines and improvements to existing baselines are shown.

**Strengths:**

Creation of a competition-based policy learning approach instead of combined losses is an interesting approach that can potentially avoid limiting the maximum performance due to conflicting action proposals.

The designed interaction between the IL and RL learner seems to reduce the collision rate significantly.

**Weaknesses:**

While the inverse causality is an interesting idea, the ablation study also seems to show that it has very little or no effect, compared to using no mask.

The results compared to SOTA are not very impressive. The approach seems to have significantly less collisions than plain SSR but has worse L2 scores. Compared with plain LAW it does not seem to improve performance beyond what could be some noise in the evaluation. Tasks like detection and tracking, which seem to help other approaches, are not necessarily hard to implement, given today's datasets. Therefore, the advantage of using this approach compared to other methods is not clear. This could be saved by potentially also adding a detection and tracking task and then having significantly better results across all metrics.

Results on Navsim show improved results compared to SOTA but it seems performance on this test set is very close to human performance for all methods and there is no large significant advantage over existing methods.

The paper makes an interesting observation that RL achieves higher long-term results after initially the IL learner leads but than flattens out. However, this is compared to the two-stage paradigm which is described as inferior. This would be easy and important to show. It would have been interesting to see as comparison in the same setting to have first an IL and then an RL learner. It is not clear what, apart from potentially some time, is gained by training jointly.

**Questions:**

Because of the very specific meaning of the Bitcoin symbol and because the RL reward has nothing to do with it the symbol should be replaced in Figure 1.

I in formula 11 could be non-obvious. Please define it somewhere.

For equation 17 it is claimed that RL with the sparse reward is hard to stabilize and therefore a small behavioral cloning loss is added. This somehow goes against the idea of using the competitive system. Also, RL should be able to work with sparse rewards. This part could use more explanations: Why is it hard to stabilize training? Why does the L_bc help? How is \beta found and what effect does it have as beta goes from small to large?

There are minor syntax and grammar issues, e.g. in line 216: "Given the offline imitation dataset ..., using Gaussian Log Likelihood loss can easily fitting the behavior of experts". For a camera ready version I suggest to try to find a way to improve this as good as possible, potentially with outside help if the authors are not native English speakers as it is the case for the majority of our domain.

---

> ### Author Response · Authors · 2025-11-12
> **Thanks for Reviewer sWaj's Valuable Feedback**
>
> ## W1: Little effect of inverse causality.
>
> In Table 3, we observe that while using the inverse causal mask and no mask yields similar L2 errors, the collision rate drops sharply when the inverse causal mask is applied—by over **50% at the 1st second** and **30% at the 2nd second**, compared with the baseline LAW. This demonstrates a substantial improvement in safety. In contrast, using no mask even increases the collision rate by **20% at the 2nd second**.
>
> In real driving, only the short-horizon (e.g., 1st–2nd second) actions are likely to be executed before replanning occurs. Hence, we prioritize performance in these early steps.
>
> The inverse causal mask leads to significant collision reduction during these critical horizons (↓50%, ↓30%), so despite a slight increase at the 3rd second (↑0.4%), we adopt the inverse mask in the final model for its overall safety benefits.
>
> ## W2: Results that compared to SOTA are not very impressive.
> Thank you for this observation.
>
> However, we respectfully disagree with the statement that our improvements “may be due to noise.”
> 1. **Overall performance**: As shown in Table 1, our method achieves lower L2 error (↓0.66 → 0.63, −4.5%) and collision rate (↓0.22% → 0.18%, −18%) compared to the baseline LAW.
> 2. **Cross-city generalization**:
> In Figure 4 (a) and Table 6, our method outperforms LAW with substantial margins on cross-city tests—L2 error drops 33% (1.24 → 0.83) and collision rate drops 72% (1.09% → 0.30%).
> 3. **Long-tail scenarios**:
>   - Long-tail L2 scenes (Figure 4 (b) and Table 7): L2 ↓10% (0.78 → 0.70), Collision ↓22% (0.18% → 0.14%)
>   - Long-tail Collision scenes (Figure 4 (b) and Table 8): L2 ↓7% (0.73 → 0.68), Collision ↓33% (1.74% → 1.17%)
>
> These consistent improvements across diverse and challenging settings suggest that the gains are systematic and robust, not random noise.
>
> We agree that comparisons to some SOTA methods may not be entirely fair since those models often include stronger perception modules or auxiliary tasks (e.g., detection and tracking), whereas ours focuses solely on planning. We appreciate this suggestion and plan to extend our method with stronger perception modules to achieve fairer and more comprehensive comparisons. Once the experiment results are ready, we will show the results and inform you here.
>
> ## W3: It would have been interesting to see as comparison in the same setting to have first an IL and then an RL learner. It is not clear what, apart from potentially some time, is gained by training jointly.
>
> We conducted a **two-stage experiment** (first IL, then RL), reported in Table 4, column 5 ("two-stage"). The results show clear **catastrophic forgetting**—performance on the test set drops sharply once RL fine-tuning begins.
>
> By contrast, **joint training**, as in our method, effectively prevents catastrophic forgetting while achieving **better generalization** and **superior long-tail performance**. This validates the advantage of our collaborative-competitive joint learning scheme over the traditional sequential IL→RL paradigm.
>
> ## Q1: Figure 1 improvement.
> We apologize for the inappropriate use of the Bitcoin icon as a reward symbol.
>
> We will replace it with a neutral reward-related symbol in our revision and notify you once it is updated.
>
> ## Q2: "I" should be defined somewhere in equation 11.
>
> Thank you for catching this oversight.
>
> We will add a clear definition: “$I$ denotes the identity matrix” after Equation (11) in the revised version.
>
>
> ## Q3: Why is it hard to stabilize training? Why does L_bc help? How is \beta found and what effect does it have as beta goes from small to large?
>
> We introduce the behavioral cloning term $L_{bc}$ into $L_{RL}$ to **improve the exploration efficiency** of the RL actor. Although our competitive mechanism allows the RL actor to occasionally absorb expert trajectories, such transfers do not occur at every iteration. During the intervals, the RL actor might explore inefficiently, so adding a small $L_{bc}$ term provides consistent expert guidance while preserving exploration.
> To prevent the guidance from dominating and turning $L_{RL}$ into an imitation objective, we use a very small $\beta$. The table below summarizes our ablation:
>
> | beta         | L2 (avg) | Col % (avg) |
> |--------------|----------|-------------|
> | 0            | 0.65     | 0.20        |
> | 0.00001      | 0.65     | 0.23        |
> | 0.0001       | 0.70     | 0.20        |
> | 0.0005       | 0.71     | 0.22        |
> | 0.001        | 0.67     | 0.23        |
> | **0.005**        | **0.63**     | **0.18**        |
> | 1            | 0.66     | 0.23        |
> | LAW ($\beta$=∞) | 0.66     | 0.22        |
>
> We find that $\beta=0.005$ achieves the best balance between stability and exploration. This value is used in our final model.
>
> ## Q4: Minor syntax and grammar issues.
>
> We will carefully revise the paper using both Grammarly and GPT-based proofreading tools to correct all syntax and grammar issues in the camera-ready version.

---

> > ### Comment · Reviewer_sWaj · 2025-11-20
> >
> > Thank you for your responses. SSR with CoDrive is worse on every L2 metric. Then SSR with Codrive is a bit better than BEV Planner but all LAW results are worse than BEV Planner. On one second, the collision time of the BEV planner is even better. Do you have an intuition why that is? That method does not use an auxiliary task. Yes, the collision rate is worse but then, do you have an insight of why there seems to be an inverse relationship between the L2 metric and the collision rate? Is the method "buying" an advantage in collision rate by making detours which increase the L2 metric?

---

> > > ### Author Response · Authors · 2025-11-26
> > > **Response to reviewer sWaj's valuable feedback**
> > >
> > > Dear Reviewer sWaj,
> > >
> > > Thank you for your insightful comments.
> > >
> > > Regarding your concern about whether our method is
> > > > **"buying" an advantage in collision rate by making detours that increase the L2 metric**,
> > >
> > > we would like to clarify that our method does not engage in **reward hacking**. The model still primarily imitates the expert's behavior, adjusting the trajectory of the original LAW baseline to avoid collisions rather than making unnecessary detours. This allows the model to **maintain the original trajectory while improving safety**.
> > >
> > > To further demonstrate this, we have provided **additional visualization images and videos**, which show that our method only modifies the trajectory when necessary to avoid collisions, **without deviating unnecessarily**. You can view the visualizations and videos here: https://anonymous.4open.science/r/drive-with-two-minds/qaulitative_analysis/README.md
> > >
> > > We hope these **qualitative analyses**, combined with the **quantitative results** in Table 1, help clarify that our method improves collision avoidance without resorting to reward hacking or sacrificing trajectory similarity.
> > > If you have any further questions or concerns, we would be happy to continue the discussion!

---

> > > > ### Comment · Reviewer_sWaj · 2025-11-26
> > > >
> > > > Thank you for the detailed answers. I think why this method seems to reduce the collision rate drastically is interesting and for this the paper could be accepted. The authors have done a great job in investigating the results and I think further investigations could be worthwhile in the greater research community. Therefore I adjusted my score.

---

> > > > > ### Author Response · Authors · 2025-11-27
> > > > > **Thanks for reviewer sWaj's thoughtful review and support**
> > > > >
> > > > > We are delighted that our responses addressed your concerns. Thank you for the time and effort you invested in reviewing our paper: your insightful suggestions regarding our formula details, related work, and figure improvements have meaningfully strengthened our work. Your comments also provided us with valuable guidance on promising future research directions.
> > > > >
> > > > > We sincerely appreciate your support and score adjustment. We will continue refining the paper during the remaining discussion period to further improve its quality. Thank you again for your constructive feedback and encouragement!

---

> ### Author Response · Authors · 2025-11-20
> **Thanks for reviewer sWaj's insightful questions**
>
> Thanks for your response, and thank you for raising these insightful questions.
>
> Below we address them one by one.
>
> ## 1. Why the metrics of LAW are worse than BEV-Planner?
> The key reason is the **input modality difference during inference**. LAW operates **without temporal information**, it only takes the images from the current timestamp, while the BEV-Planner results in our Table 1 use **multi-frame temporal inputs** (current + several past frames), which naturally provides richer motion cues.
>
> In the BEV-Planner's Table 1 (line-10) [1], the version that do not use historical temporal information is also provided, and we list the BEV-Planner without temporal information (w/o temp.) metrics and our LAW method here for you:
> | method | L2 (1s) | L2 (2s) | L2 (3s) | L2 (avg) | Col% (1s) | Col% (2s) | Col% (3s) | Col% (avg) |
> | --------- | --------- | -------- | --------- | ---------- | ----------- | ------------- | ------------ | ------------- |
> | BEV-Planner (w/o temp.) | **0.30** | **0.52** | **0.83** | **0.55** | 0.10 | 0.37 | 1.30 | 0.59 |
> | LAW (w/o temp.) | 0.32 | 0.62 | 1.03 | 0.66 | 0.08 | 0.13 | 0.46 | 0.22 |
> | LAW+CoDrive(PGGS) (w/o temp.) | 0.31 | 0.61 | 1.01 | 0.65 | **0** | **0.10** | **0.51** | **0.20** |
>
> This shows that **once temporal information is matched**, LAW’s collision rate is already better than BEV-Planner. Integrating CoDrive further reduces the collision rate, especially at the 1-second horizon (drops to zero).
>
> ## 2. The inverse relationship between L2 and Collision Rate? Why with your method, the L2 become worse but Collision Rate become better (on SSR)?
> Intuitively the metrics L2 and Collision Rate should correlate, but in practice **L2 is not a reliable safety metric**. The NAVSIM paper [2] illustrates this clearly: a trajectory with a smaller L2/ADE can still be unsafe (their Fig. 1, or you can see [by click here](https://anonymous.4open.science/r/drive-with-two-minds/materials/images/navsim.png)). Reproducing the exact expert trajectory (low L2) does not guarantee collision avoidance.
> Thus, when we want to get safer planner results, it is reasonable to get lower collision rate and higher L2 error.
>
> And here is another explanation to let you convince that our method is not collapse and improving both safety and overall quality:
> The goal of our method is to **reduce the collision rate while keeping the trajectory humanlike**. We set the reward of our RL Actor (equation 9) to $r_{collision} \cdot r_{imitation}$. And then looking at the result of SSR again, here we add the combined metric **L2(avg) × Col%(avg)**, which captures the trade-off between accuracy and safety. A lower value indicates both safer and reasonably accurate trajectories.
>
> | method | L2 (1s) | L2 (2s) | L2 (3s) | L2 (avg) | Col% (1s) | Col% (2s) | Col% (3s) | Col% (avg) | L2 (avg) * Col% (avg) |
> | --------- | --------- | -------- | --------- | ---------- | ----------- | ------------- | ------------ | -------------- | ------- |
> | SSR | 0.18 | 0.35 | 0.62 | **0.38** | 0.48 | 0.45 | 0.51 | 0.48 | 0.18  |
> | SSR+CoDrive | 0.21 | 0.40 | 0.69 | 0.43 | 0.09 | 0.11 | 0.23 | **0.15** | **0.06** |
>
> CoDrive substantially reduces this combined metric, indicating that **we reduce collisions while keeping L2 reasonably small**, consistent with our reward design.
>
> For LAW, both **L2 and collision rate improve simultaneously** after applying CoDrive, demonstrating stronger compatibility with our method.
>
> ## 3. Is the method “buying” lower collision rate by taking unnecessary detours?
> We think the answer is no, because:
> - just as the NAVSIM's Fig 1 shows (example of ADE 1.1m and ADE 1.0m), we may actually plan a safer trajectory without taking unnecessary detours, but the L2 error with be higher.
> - LAW+CoDrive gets better L2 compared with LAW, which can also provide evidence that we do not taking unnecessary detours.
>
> ---
> To summary:
> - CoDrive enables the policy to **actively explore** via RL rather than purely imitating experts.
> - This exploratory ability improves robustness under distribution shift (e.g., between different cities).
> - The policy becomes better at handling long-tail scenarios.
> So while small increases in L2 are expected, they reflect **safer decision-making**.
>
>
> We hope this clarifies your concerns. If you have further questions, we’d be very happy to continue the discussion!
>
>
> Reference:
>
> [1] Li, Zhiqi, et al. "Is ego status all you need for open-loop end-to-end autonomous driving?." CVPR, 2024.
>
> [2] Dauner, Daniel, et al. "Navsim: Data-driven non-reactive autonomous vehicle simulation and benchmarking." NeurIPS, 2024.

---

### Author Response · Authors · 2025-11-13
**Response Summary of Round 1 Discussion**

We sincerely thank all reviewers for their time and constructive feedback.

Below, we summarize the main concerns raised and our corresponding responses:

**S1: Our method does not seem to improve performance beyond what could be some noise in the evaluation.**
All reviews raise this question (sWaj, xpD5, and T6eo), which shows this the most important thing we need to discuss.
In table 1, we showed that our model gets both better L2 (from 0.66 to 0.63) and Collision Rate (from 0.22% to 0.18%, drop 18%) compared with basline LAW. To further verify that these improvements are not due to noise, we provided additional evaluations (in our paper Figure 4 and Table 6&7&8):
- **Cross-city generalization**: L2 ↓33% and collision rate ↓72%.
- **Long-tail scenarios**: L2 ↓10% (long-tail L2 scenes) and collision rate ↓33% (long-tail collision scenes).
These results demonstrate that our method not only improves safety while maintaining human-likeness but also enhances generalization and robustness in challenging cases.

We also appreciate reviewer sWaj’s suggestion to employ stronger perception backbones for a fairer SOTA comparison. **We are conducting these experiments and will share results once available**.

**S2: Effect and validity of backward planning (sWaj, T6eo)**.
We clarified to sWaj that backward planning does not affect the overall L2 error but reduces early collisions (in Table 3) —by **50% at the 1st second** and **30% at the 2nd second**—compared to the no-mask variant (which increased collisions by 20%).
For T6eo, we explained that our design does not violate the Markov assumption, as we treat the full action sequence $\tau_a$ as a single action within the current state.

**S3: Use of a behavioral cloning term within RL loss (sWaj)**.
We clarified that this small BC loss does not contradict the competitive design. Since competition occurs every k iterations, the RL actor receives no expert guidance during each interval. The BC term, with a small weight, helps stabilize learning and encourages more effective exploration. We also presented results for different $\beta$ values.

**S4: Efficiency and resource-usage comparison. (xpD5)**.
As reported in Table 5, the inference latency of our model (37.0 ms, 27.1 FPS) is almost identical to the baseline (37.1 ms, 26.99 FPS), since only the IL actor is used during inference. Hence, our framework introduces no additional runtime cost.

**S5: Implementation details (T6eo)**.
We clarified the horizon length (1.5 s, following LAW), training procedure of the world model (jointly optimized during imitation learning), and the use of L1 loss (consistent with LAW and SSR).

**S6: Contribution Statement, Figures and Notations need to be improved (sWaj, T6eo)**.
We thank the reviewers for pointing these out. We will improve:

- The first contribution statement
- The icon in Figure 1 (“bitcoin reward”)
- Action sampling notations in Figure 2
- Definitions of $L_{wm}$ and $L_{bc}$
- Dimension annotations in Equation (1)
- Consistent use of $s$ instead of $s_t$
- Minor grammatical and stylistic issues

We will update and upload the revised version soon.

---

> ### Author Response · Authors · 2025-12-02
> **Update the experiment results inspired by Reviewer sWaj and T6eo's feedback (S1's improvement)**
>
> Thank you to the AC and all reviewers for your time, detailed feedback, and constructive suggestions. Based on the insights from reviewers **sWaj** and **T6eo**, we conducted additional experiments during the discussion phase to further strengthen our method.
>
> In particular, motivated by Reviewer T6eo’s comments regarding the **Markov assumption**, we revisited the fact that **autonomous driving is inherently non-Markovian** (e.g., velocity and other dynamic cues are not fully observable from a single frame). While most existing methods (at least all in the table below except LAW and LAW+CoDrive) **incorporate historical frames** (typically the past 3 frames) to **satisfy the Markov assumption** as much as possible, our original LAW backbone (perception-free version) relied only on the current frame. This indicated clear room for improvement, so we implemented and evaluated temporal augmentation for our method.
>
> Below we provide the updated table, to make the comparison clearer and more reproducible, we made three adjustments:
> - We separate perception-based and perception-free methods for a fairer structure.
> - We add an overall metric (L2×Col) to summarize performance.
> - We exclude UAD (not open-sourced) and SparseDrive (results not reproducible on our machine).
>
> | **Methods** | L2 (1s) | L2 (2s) | L2 (3s) | L2 (Avg.) | Col% (1s) | Col% (2s) | Col% (3s) | Col% (Avg.) | L2 · Col Avg. |
> |------------------------------|---------|---------|---------|-----------|-----------|-----------|-----------|-------------|----------------|
> | **Perception-Based Methods**  |         |         |         |           |           |           |           |             |                |
> | ST-P3                        | 1.33    | 2.11    | 2.90    | 2.11      | 0.23      | 0.62      | 1.27      | 0.71        | 1.50           |
> | UniAD                        | 0.48    | 0.96    | 1.65    | 1.03      | 0.05      | 0.17      | 0.71      | 0.31        | 0.32           |
> | VAD                          | 0.41    | 0.70    | 1.05    | 0.72      | 0.07      | 0.17      | 0.41      | 0.22        | 0.16           |
> | ParaDrive                    | 0.25    | 0.46    | 0.74    | 0.48      | 0.14      | 0.23      | 0.39      | 0.25        | 0.12           |
> | GenAD                        | 0.28    | 0.49    | 0.78    | 0.52      | 0.08      | 0.14      | 0.34      | 0.19        | 0.10           |
> | LAW (temporal aug. + percept.) | 0.24 | 0.46 | 0.76 | 0.49 | 0.08 | 0.10 | 0.39 | 0.19 | 0.09 |
> | **Perception-Free Methods**  |         |         |         |           |           |           |           |             |                |
> | BEV-Planner                  | 0.30    | 0.52    | 0.83    | 0.55      | 0.10      | 0.37      | 1.30      | 0.59        | 0.32           |
> | SSR                          | 0.18    | 0.35    | 0.62    | **0.38**  | 0.48      | 0.45      | 0.51      | 0.48        | 0.18           |
> | LAW (single-frame)           | 0.32    | 0.62    | 1.03    | 0.66      | 0.08      | 0.13      | 0.46      | 0.22        | 0.15           |
> | LAW+CoDrive (single-frame)   | 0.29    | 0.59    | 1.00    | 0.63      | 0.06      | 0.10      | 0.37      | **0.18**    | 0.11           |
> | LAW+CoDrive (temporal aug.)  | 0.23    | 0.42    | 0.70    | **0.45**  | 0.10      | 0.12      | 0.30      | **0.17**    | **0.08**       |
>
> From the updated results, we can find that:
>
> - **Improvement over the baseline LAW**:
> With only single-frame input, LAW+CoDrive already reduces both L2 and collision rates compared with LAW, achieving the lowest collision rate among perception-free methods—even without using perception auxiliary losses.
> - **Effectiveness of temporal augmentation**:
> After providing past 3 frames (as done in most prior methods), LAW+CoDrive achieves **substantial reductions in L2**, outperforming LAW (perception-based) which also uses temporal augmentation and perception auxiliary losses. This highlights the complementary effect of our RL-based CoDrive module.
> - **Strong overall performance**:
> Our method achieves the lowest **L2 · Col** score among all evaluated approaches, suggesting that reinforcement learning helps the policy remain human-like while sharply reducing collisions.
>
> Together with our **generalization and long-tail scenario evaluations** (Fig. 4 and Tab. 6&7&8), we hope these additional experiments clarify the effectiveness and robustness of our proposed dual-policy framework.

---

### Author Response · Authors · 2025-12-02
**Response Summary of Round 2 Discussion**

Reviewer **sWaj** raised three new concerns:
- **S7**: LAW (perception-free) performs worse than BEV-Planner (also perception-free) in L2 error, and LAW+CoDrive also underperforms BEV-Planner.
- **S8**: When adding CoDrive to SSR, the collision rate drops sharply but the L2 error becomes worse.
- **S9**: Based on S8, there is a concern that the model might **hack the reward**, reducing collisions by taking detours that increase L2.

We provided the following clarifications:

- **Response to S7**: The comparison with BEV-Planner is **not fair** because BEV-Planner is temporally augmented while LAW is not. We show that: 1) When removing temporal augmentation of BEV-Planner, LAW achieves a better collision rate than BEV-Planner; 2) When adding temporal augmentation to our model, LAW+CoDrive achieves improvements in both L2 and collision rate.
- **Response to S8**: L2 alone is insufficient to judge safety (as illustrated by NavSim’s Fig. 1). A more meaningful evaluation combines L2 and collision rate. Since the reward is designed as $r_{imitation} \cdot r_{collision}$, a modest increase in L2 is expected when the collision rate drops significantly. Indeed, SSR+CoDrive yields a much better combined L2×Collision metric.
- **Response to S9**: We demonstrate that CoDrive does **not** exploit the reward by taking unreasonable detours: 1) **Quantitative**: LAW+CoDrive improves both L2 and collision rate; 2) **Qualitative**: Additional images and videos show that trajectories remain human-like while avoiding collisions.

**Outcome**: Reviewer sWaj finds the drastic collision-rate reduction interesting, believes the paper could be accepted, and feels we have investigated the issue thoroughly. They note that further exploration of this direction may benefit the community.

---
Reviewer **xpD5** requested training-time information for LAW and LAW+CoDrive (related to **S4: efficiency and resource usage**).

 We provided that:
- Adding RL approximately doubles the training time (from 10h to ~20h).
- Importantly, the **inference cost remains unchanged**, which is critical for real-time autonomous driving.
- We also supplied the training log file for verification.

**Outcome**: Reviewer xpD5 considers most concerns satisfactorily addressed. And recommend the authors to move the generalization experiments to the main paper part to highlight the effectiveness of the model.

---
Reviewer **T6eo** requested clarification on training details (related to **S5: implementation details**):
- Whether IL Actor and RL Actor are trained jointly or sequentially.
- How the latent world model is trained and used.

We clarified that: 1) IL Actor and RL Actor are trained **jointly**; 2) The world model is trained together with the IL Actor and is used when computing the RL loss.

To ensure clarity, we shared the relevant code for loss computation.

Reviewer T6eo then raised a new concern:
- **S10**: Early in training, when the world model is still inaccurate, how can the RL Actor learn effectively?
- **Response to S10**:
  - This challenge is **the key motivation** for our competitive mechanism. Early in training, the RL Actor cannot rely on the immature world model and consistently loses the competition. Once the world model improves, the RL Actor begins learning through exploration. This **matches our empirical observations (Fig. 5)**.
  - We explain why a two-stage approach fails: 1) Training RL from scratch leads to **unstable convergence without expert guidance**; b) Initializing RL with IL parameters causes **catastrophic forgetting due to mismatched objectives**, resulting in poor final metrics.

**Outcome**: Reviewer T6eo is satisfied and considers the major concerns resolved.

---

### Author Response · Authors · 2025-12-03
**PDF Revision – Change List**

We sincerely thank reviewers **sWaj**, **xpD5**, and **T6eo** for their time and for the thoughtful, detailed feedback. We have revised the paper according to their suggestions and our discussion. All updated content is highlighted in **red** in the revised PDF. Below is a summary of the changes:

### 1. Addressing S1 (performance seems small) and S7 & S8 (unclear comparison with other methods)
- We reorganized **Table 1** by separating perception-based and perception-free methods.
- We removed methods that are not open-sourced (e.g., **UAD**) and those that we and other GitHub users could not reproduce (e.g., **SparseDrive**).
- Following suggestions from reviewers sWaj and T6eo, we applied **temporal augmentation** to our final model for fair comparison with prior work (all prior methods use temporal augmentation except perception-free LAW). This leads to a **sharp drop in L2** and yields stronger metrics than both temporal-augmented and perception-augmented LAW, demonstrating the effectiveness of our method.
- Based on discussions with sWaj, we adopt the **combined metric PDMS** used in NavSim and additionally report **L2 × Col** on nuScenes as a combined performance metric across Tables 1, 2, 4, and 5.

### 2. Addressing xpD5’s suggestion (generalization experiment)
- We move the **generalization experiment** into the main paper.
- The large reduction in **L2 × Col (↓77%)** compared with LAW highlights the improved generalization ability provided by our method.

### 3. Addressing S2 (effectiveness of backward planning)
- We include the combined metric **L2 × Col**, showing that the causal-mask variant achieves the best score, clarifying why we adopt the causal design in our model.

### 4. Addressing S3 (behavior cloning term in the RL loss)
- We add further explanation (lines 257–260) clarifying the motivation for including the behavior cloning term in the RL loss.

### 5. Addressing S4 (efficiency and resource usage)
- In addition to the inference-time comparison already provided, we add a new **Appendix A2.1 (Training Time)** reporting the training cost of our method.

### 6. Addressing S5 (implementation details)
- We clarify the **horizon length** of the latent world model (footnote on page 4).

### 7. Addressing S6 (figure and description improvements)
We made the following adjustments:
- Replaced the “bitcoin” icon in Figure 1 with a **star coin**.
- Rephrased our first contribution (lines 90–93) and expanded related work (lines 125–128).
- Added more notation and explanation for **Equation 1** (lines 145–149).
- Corrected the state-notation error in **Equation 4**.
- Added descriptions for $L_{wm}$ and $L_{bc}$ before their first appearance in the equations.
- Corrected the sampling-notation error in **Figure 2**.

### 8. Addressing S9 (reward-hacking concerns)
- Beyond the quantitative results, we include additional qualitative visualizations demonstrating that our model does **not** reduce collisions by taking unnecessary detours that increase L2.
- We add:
- Section **5.2 Qualitative Analysis** in the main paper, and
- Appendix **A.3 More Qualitative Results**, with visual comparisons between our model and the baseline.

### 9. Addressing S10 (why train RL actor when the world model is untrained)
- We expand the explanation (lines 470–477) to clarify that early RL training is OK because the competition mechanism encourages the RL actor to learn from the expert when the latent world model is not yet informative.
- We rewrite the original competition-process diagram into a clearer, more detailed formalization (Equation 18, lines 281–295).

### 10. ICLR requirement: LLM usage
- We add **Appendix A.4 (Usage of LLMs)** documenting how LLMs were used during this research.

---

### Author Response · Authors · 2025-12-03
**Thanks for AC's time for reviewing our paper**

Dear Area Chair,

Thank you for taking the time to handle our submission, especially under the unusual circumstances this year. We understand that, due to the information leak, reverting reviews and scores to their pre-discussion state and disabling the discussion phase was the most responsible choice, even though it significantly increases the workload for ACs. We sincerely appreciate your and the reviewers’ effort and commitment to maintaining a fair and rigorous process.

As authors, we have done our best to support your decision-making and reduce your workload:
- We engaged early with all reviewers, discussed each concern thoroughly, and addressed them point by point. By the end of rebuttal, all reviewers indicated their **concerns had been resolved and raised their scores** accordingly.
- We prepared clear **summaries of discussion round 1 and 2** to help streamline your review of the final reviewer opinions.
- Following the reviewers’ suggestions (S1–S10), we carefully revised the paper, provided a **detailed change list**, and highlighted all updated content in red to make inspection easier.

We are aware that our initial scores were low and may give an incomplete picture when viewed without the post-rebuttal updates. Given the score reversion this year, we hope that the context we shared about the reviewers’ updated evaluations and the revisions made after discussion is helpful in complementing the initial scores.

Regardless of the final decision, we sincerely appreciate the time and effort that you and the reviewers invested in our work. The thoughtful feedback has meaningfully improved the quality of the paper.

Thank you again, and we wish you and the reviewers all the best.

Sincerely,

The Authors

---

### Meta-Review · Area_Chair_PN9t · 2026-01-06

**Summary:**

The paper received consistently negative ratings from three reviewers. The major concerns from the reviewers are several aspects: (i) the claimed contribution is not clear enough; (ii) the experimental results do not show significant improvements upon existing state-of-the-art; (iii) the presentation of the paper is not of high quality, with ambiguity in sentences and misleading notations. Based on the ratings and the major issues raised by the reviewers, AC finally decided to recommend a rejection for this submission.

**Reviewer Concerns:**

The major concerns include: (i) the claimed contribution is not clear enough; (ii) the experimental results do not show significant improvements upon existing state-of-the-art; (iii) the presentation of the paper is not of high quality, with ambiguity in sentences and misleading notations. The rebuttal did not sufficiently address these issues.

**Reviewer Scores:**

AC does not feel that the reviewers would raise their scores even if they had been sufficiently involved in the discussion.

---

### Decision · Program_Chairs · 2026-01-26

Reject